# Validation, comparison, and integration of GOCI, AHI, MODIS, MISR, and VIIRS aerosol optical depth over East Asia during the 2016 KORUS-AQ campaign

Myungje Choi[1,2], Hyunkwang Lim[2], Jhoon Kim[2], Seoyoung Lee[2], Thomas F. Eck[3, 4], Brent N. Holben[4],
Michael J. Garay[1], Edward J. Hyer[5], Pablo E. Saide[6], Hongqing Liu[7, 8]

[1]Jet Propulsion Laboratory, California Institute of Technology, Pasadena, California, USA
[2]Department of Atmospheric Sciences, Yonsei University, Seoul, Republic of Korea
[3]Universities Space Research Association, Columbia, MD, USA
[4]NASA Goddard Space Flight Center, Greenbelt, MD, USA
[5]Marine Meteorology Division, Naval Research Laboratory, Monterey, CA, USA
[6]Department of Atmospheric and Oceanic Sciences, Institute of the Environment and Sustainability, University of California–Los Angeles, Los Angeles, CA, USA
[7]National Oceanic and Atmospheric Administration, National Environmental Satellite, Data, and Information Service, Center
for Satellite Applications and Research, College Park, MD, USA
[8]I. M. Systems Group, Inc., College Park, MD, USA

*Correspondence to*: Jhoon Kim (jkim2@yonsei.ac.kr); Myungje Choi (myungje.choi@jpl.nasa.gov)

## Abstract

Recently launched multi-channel geostationary-Earth-orbit (GEO) satellite sensors such as the Geostationary Ocean Color Imager (GOCI) and the Advanced Himawari Imager (AHI) provide aerosol products over East Asia with high accuracy, which enables the monitoring of rapid diurnal variations and the transboundary transport of aerosols. Most aerosol studies to date have used low-Earth-orbit (LEO) satellite sensors, such as the Moderate Resolution Imaging Spectroradiometer (MODIS) and the Multi-angle Imaging SpectroRadiometer (MISR) with a maximum of one or two overpass daylight times per day at mid-
to low latitudes. Thus, the demand for new GEO observations with high temporal resolution and improved accuracy has been significant. In this study the latest versions of aerosol optical depth (AOD) products from three LEO sensors—MODIS (Dark-Target, Deep-Blue, and MAIAC), MISR, and the Visible/Infrared Imager Radiometer Suite (VIIRS)—along with two GEO sensors—GOCI and AHI—are validated, compared and integrated for the period during the Korea–United Sates Air Quality Study (KORUS-AQ) field campaign from 1 May to 12 June 2016 over East Asia. The AOD products analyzed here generally
have high accuracy with high *R* (0.84–0.93) and low *RMSE* (0.12–0.17) but their error characteristics differ according to the use of several different surface-reflectance estimation methods. High-accuracy near-real-time GOCI and AHI measurements facilitate the detection of rapid AOD changes, such as smoke aerosol transport from Russia to Japan on 18–21 May 2016, heavy pollution transport from China to Korea on 25 May 2016, and local emission transport from the Seoul Metropolitan Area to the Yellow Sea in Korea on 5 June 2016. These high-temporal-resolution GEO measurements result in more-

representative daily AOD values and make a greater contribution to a combined daily AOD product assembled by median-value selection with a 0.5° × 0.5° grid resolution. The combined AOD is spatially continuous and a greater number of pixels with high accuracy (fraction within expected error range of 0.61) than individual products. This study characterizes aerosol measurements from LEO and GEO satellites currently in operation over East Asia, and results presented here can be used to
evaluate satellite measurement bias and air-quality models.

## 1 Introduction

Atmospheric aerosol particles are composed of solid and liquid matter and have diameters of a few nanometers up to several micrometers and lifetimes of one to tens of days. Aerosol particles affect the atmospheric radiation balance by scattering and absorbing incident top-of-atmosphere (TOA) sunlight and that scattered from the surface, as well as by interacting with clouds
(e.g., by changing cloud distributions, optical properties, and precipitation by acting as cloud condensation nuclei) with global climate effects (IPCC, 2013). Global net radiative cooling or heating is determined partially by interactions for which the level of understanding is still low and varies significantly with geographic region. Additionally, ambient particulate matter (PM) at the ground level adversely affects human health through pulmonary and respiratory transport, resulting in heart disease, stroke, and lung cancer (Gao et al., 2015; Lim et al., 2012). Many developing countries in East Asia have both large anthropogenic
emission sources and natural aerosol sources, such as the Taklamakan and Gobi deserts and wild-fire regions. For this reason, East Asia currently has one of the most heavily polluted atmospheres in the world (S. W. Kim et al., 2007; Mehta et al., 2016; Yoon et al., 2014; Zhao et al., 2017).

Aerosol measurements are routinely conducted at diverse scales by laboratory experiments, in situ measurements and remote sensing, and from various platforms including ground-based, airborne, ship-borne, and satellite sensors. Accurate
microphysical and chemical properties of aerosols can be obtained from laboratory experiments or ground-based/airborne measurements, but their spatial coverage is limited. Satellite-based remote sensing measurements provide aerosol optical properties, including aerosol optical depth (AOD), over much broader areas. Currently operating low-Earth-orbit (LEO) satellite sensors such as the Moderate Resolution Imaging Spectroradiometer (MODIS), the Multi-angle Imaging SpectroRadiometer (MISR), and the Visible/Infrared Imager Radiometer Suite (VIIRS) provide global aerosol information,
but at a temporal resolution that is limited to once per day at least, but typically once every 2–3 days due to cloud cover (Garay et al., 2017; Hsu et al., 2013; Jackson et al., 2013; Levy et al., 2013). Most satellite-based aerosol retrieval techniques and algorithms have been developed for these LEO sensors (Diner et al., 1998; Higurashi and Nakajima, 1999; Hsu et al., 2004; Kaufman et al., 1997; J. Kim et al., 2007; Remer et al., 2005; Torres et al., 1998). To overcome temporal resolution limitation, there were several attempts to retrieve AOD using first-generation meteorological geostationary satellites such as the
Geostationary Operational Environmental Satellite (GOES), the Geostationary Meteorological Satellite (GMS), and the Multifunction Transport Satellite (MTSAT), but they showed worse accuracy than those of LEO sensors due to the wider and fewer visible channels with coarser spatial resolution which make difficult to distinguish aerosol types (Kim et al., 2008;

Knapp et al., 2002; Urm and Sohn, 2005; Wang et al., 2003; Yoon et al., 2007). As the specifications of recently launched geostationary-Earth-orbit (GEO) sensors such as the Geostationary Ocean Color Imager (GOCI) and the Advanced Himawari Imager (AHI) over East Asia and the Advanced Baseline Imager (ABI) over the United States are approaching those of current LEO sensors, aerosol optical properties can be retrieved with accuracy as high as that of LEO sensors, and at much higher temporal resolutions, from a few minutes to one hour during daylight hours (Chen et al., 2018; Choi et al., 2018; Choi et al., 2016; Daisaku, 2016; Kikuchi et al., 2018; Laszlo and Liu, 2016; Lee et al., 2010; Lim et al., 2018; Zhang et al., 2018). This breakthrough in temporal resolution of GEO aerosol data enables us to monitor highly variable aerosol conditions and improve air-quality forecasting, particularly for PM, with data assimilation (Jeon et al., 2016; Lee et al., 2016; Pang et al., 2018; Park et al., 2014; Saide et al., 2014) or machine learning (Park et al., 2018). To improve air quality model accuracy through satellite AOD retrieval, the satellite AOD should have broader coverage, high spatiotemporal resolution, and high accuracy. Most AOD data assimilation system has been developed by using LEO satellite products such as MODIS because they have global coverage and high accuracy through the continuous retrieval algorithm improvement. The GEO satellite can provide more frequent AOD, but its spatial coverage can be limited to the specific area, especially in the case of GOCI. The period of AOD retrieval algorithm development and investigation using GEO is relatively shorter than LEO. Also, generally LEO sensors have more suitable channels with high resolution and advanced measurement characteristics such as multi-angle and/or polarization for aerosol retrievals, which could result in higher accuracy of AOD from LEO than GEO generally (Jiang et al., 2019). Therefore, both accuracy and spatiotemporal coverage can be obtained simultaneously by using combined GEO and LEO AODs. For these reasons, the demand for GEO aerosol measurements is high.

Satellite aerosol retrieval algorithms have been improved toward higher spatial resolution (e.g. 5-10 km or finer), higher temporal resolution (e.g. from daily to hourly and few minutes resolution) with higher accuracy (e.g. AOD uncertainty less than 0.03 or 10%) to fulfill the requirement for understanding of long-term climatological changes (GCOS, 2016).

Several field campaigns have been performed over East Asia to investigate aerosol chemical, microphysical, and optical properties based on in situ and remote-sensing measurements. These include the Transport and Chemical Evolution over the Pacific (TRACE-P) aircraft campaign in 2001 (Jacob et al., 2003), the Atmospheric Brown Cloud–East Asia Regional Experiment (ABC–EARES) in 2005 (Nakajima et al., 2007), the Distributed Regional Aerosol Gridded Observation Networks (DRAGON)–Asia campaign in 2012 (Holben et al., 2018), and the Megacity Air Pollution Studies (MAPS) in 2015 (Kim et al., 2018). Aerosol retrieval algorithms have been developed, improved, and validated using the extensive measurement datasets obtained from these field campaign studies (Garay et al., 2017; Jeong et al., 2016; Kim et al., 2016; S. W. Kim et al., 2007; Lee et al., 2018; Xiao et al., 2016).

The Korea–United Sates Air Quality Study (KORUS-AQ; https://www-air.larc.nasa.gov/missions/korus-aq/) was performed over Korea from 1 May to 12 June 2016 under the leadership of Korea's National Institute of Environmental Research (NIER) and the United States National Aeronautics and Space Administration (NASA). Compared to previous field campaigns the KORUS-AQ campaign consists of more extensive measurement platforms from ground sites, aircrafts, ship, and satellites, especially focused on geostationary satellites. Korea have a capability of geostationary satellite aerosol measurement using

GOCI as near-real-time (i.e. within 1 hour after measurement) for the campaign and it is also an optimized testbed of future geostationary air quality satellites of GEMS (Geostationary Environment Monitoring Spectrometer; Kim et al., 2019), TEMPO (Tropospheric Emissions: Monitoring of Pollution; Zoogman et al., 2017), and Sentinel-4 (Ingmann et al., 2012). During the campaign, GOCI aerosol optical properties were retrieved and provided as near-real-time to support air-quality forecasting,

determination of the flight plan for aircraft measurements to detect heavy pollution plumes, and data assimilation using near-real-time chemical transport model simulations (Choi et al., 2018; Saide et al., 2014). Also, the GOCI AOD was used to evaluate a Japanese non-hydrate static icosahedral atmospheric model (NICAM) AOD and to analyze diurnal variation of AOD and $PM_{2.5}$ over Korea (Goto et al., 2019; Lennartson et al., 2018). Since most aerosol analysis and application studies using satellite still have been demonstrated by using LEO satellite products such as MODIS, new aerosol products from GEO

should be verified in terms of quality and spatiotemporal coverages through comparison with LEO satellite measurement. Therefore, this study aims to validate multiple GEO and LEO satellite AOD products using ground-based AERONET, compare specific aerosol event cases to understand different characteristics of GEO and LEO AODs, and integrate them for representative AOD distribution.

Here, in addition to the GOCI AOD dataset, the latest versions of AOD datasets from LEO sensors (MODIS, MISR, and VIIRS)

and another GEO sensor (AHI) are validated, compared, and integrated for the period of the field campaign. The latest version of the Aerosol Robotic Network (AERONET) of ground-based sun photometers dataset (Version 3) over East Asia is used as a reference for the campaign period (Eck et al., 2018; Giles et al., 2019). Characteristics of the various AOD products are analyzed for specific transport cases with high temporal resolution at the daily scale over the campaign period.

The remainder of this paper is organized as follows. In Sect. 2, satellite and ground-based remote sensing data used in this

study are summarized. In Sect. 3, the various AOD products are validated and compared using ground-based AERONET observations separately over ocean and land. The specific aerosol loading cases during the campaign are analyzed in Sect. 4. In Sect. 5, daily representative AOD is generated on a common spatial grid for each product and used to calculate the mean AOD distribution during the campaign. The daily AOD integration is tested using multiple AOD products at the daily scale. Finally, a discussion and conclusions are presented in Sect. 6.

## 25   2 Satellite and ground-based AERONET aerosol data

### 2.1 GOCI Yonsei aerosol product

The GOCI is a unique ocean color sensor in GEO (longitude 128.2°E) on board of the Korean Communication, Ocean, and Meteorological Satellite (COMS) and has been making observations over East Asia since 2010. It covers a 2500 km × 2500 km area centered over the Korean Peninsula, such that the eastern part of China and Japan are also covered at a 500 m × 500

m spatial resolution and 1 h temporal resolution from 09:30 to 16:30 local time (total of 8 measurements during daylight hours). Aerosol optical depth at 550 nm is retrieved using the GOCI Yonsei aerosol retrieval (YAER) algorithm at a 6 km × 6 km spatial resolution after masking pixels affected by clouds or sun-glint and aggregating the remaining pixels to provide

aerosol signals at the resolution of the reflectance measurements. Land-surface reflectance is obtained using the minimum reflectance technique for each month and hour of Rayleigh-corrected reflectance measurements (Herman and Celarier, 1997; Hsu et al., 2004; Koelemeijer et al., 2003). Ocean-surface reflectance is based on the Cox–Munk ocean bidirectional reflectance distribution function (BRDF) (Cox and Munk, 1954). Details of the aerosol retrieval algorithm, and the

improvement and validation results for March 2011 to February 2017, are presented by Choi et al. (2016). According to Choi et al. (2018), the GOCI YAER version 2 AOD shows increased errors when the geometrical cloud fraction within AOD pixel increases (particularly near cloud edges) and the remaining cloud contamination was largely due to the absence of infrared (IR) measurement in GOCI. Thus, in this study, additional cloud masking is applied to the GOCI cloud-masking procedure, using Himawari-8 IR data. The IR cloud masking processes in the AHI YAER algorithm is

summarized as in Table 2, which are based on several previous studies (Iwabuchi et al., 2014; Kim et al., 2014). It consists of four tests using brightness temperature (BT) difference (BTD) of different IR channel pair to detect high-level cloud, low-level cloud, and cirrus cloud which are difficult to be detected or classified using only visible channels of GOCI. The AHI IR cloud masking information has 2 km spatial resolution (at equator) at every 10 minutes for full-disk area, thus the collocation processing is required to match with GOCI of 500 m spatial resolution and 1-hour temporal resolution. The spatially closest

AHI IR pixel with each GOCI pixel is collocated. The GOCI observation takes from 15 to 45 minutes at every hour, thus the pixels are flagged as cloud if at least one of 10, 20, 30, 40, and 50 minute at each hour when AHI measurements determine the pixels as cloud. Then, the collocated AHI IR cloud information is applied to the GOCI at 500 m spatial resolution between the original cloud masking step and the pixel aggregation step to 6 km. Details of full AHI IR cloud masking procedure for aerosol retrieval are described by Lim et al. (2018). On 22 May 2016, thin cloud was detected over Manchuria

(125-140°E and 44-48°N) and southern ocean of Japan (128-140°E and 30-34°N) from GOCI true color image (Fig. 1a). Original cloud masking of GOCI algorithm couldn't classify pixels as cloudy near cloud edge and result in high AOD close to 2 (Fig. 1b). Those pixels are well filtered out when AHI IR cloud masking was applied to the GOCI algorithm (Fig. 1c). The GOCI YAER V2 AOD before and after additional AHI IR cloud masking was also compared to the ground-based AERONET in section 3.

**2.2 AHI Yonsei aerosol product**

The AHI, onboard the Himawari-8 and -9 satellites, is part of a new generation of meteorological satellite sensors. Compared with previous meteorological sensors, such as the Japanese Advanced Meteorological Imager (JAMI) onboard the Japanese Multifunction Transport Satellite–1R (MTSAT–1R; also referred to as Himawari-6) or the Meteorological Imager (MI) onboard the Korean COMS satellite, the AHI has more channels (16) including 3 visible channels (0.47, 0.51, and 0.64 μm)

with higher spatial resolution (0.5 to 2.0 km). The addition of more visible channels to the one broad visible channel of JAMI and MI enables aerosol type classification and improves aerosol retrieval accuracy. However, the primary advantage of AHI is the high temporal resolution: 2.5 min over Japan and 10 min over the full-disk area centered at 140.7°E. The full-disk observation area covers East Asia (eastern India, Southeast Asia, Korea, Japan, most of China, and parts of Russia and

Oceania). The AHI Yonsei aerosol retrieval algorithm applies two distinct schemes (Lim et al., 2018). The first scheme is based on the Dark-Target approach over land using a 1.6 μm short-wave infrared (SWIR) channel and the Cox–Munk ocean BRDF model with chlorophyll-*a* to simulate the water-leaving radiance over the ocean, referred to as the Estimated Surface Reflectance (ESR) method. Official AHI JAXA chlorophyll-*a* concentration data are spatiotemporally interpolated to the

AHI Yonsei aerosol product pixels and used to calculate the ocean-surface reflectance. The second scheme is based on the minimum surface reflectance over land and ocean, referred to as the Minimum Reflectance Method (MRM). The AHI YAER algorithm provides two versions of 550 nm AOD with a 6 km × 6 km spatial resolution. Details of the AHI YAER algorithm are presented by Lim et al. (2018). The full-disk area is measured by AHI in ten segments from north to south. The 2nd and 3rd segments approximately cover the area 20°N–48°N and are used to retrieve AOD using the YAER algorithm in

this study.

## 2.3 MODIS Dark-Target aerosol product

The MODIS is one of the most widely used instruments for global aerosol measurements. It has been in operation onboard the NASA Terra (10:30 descending) satellite since 1999 and the Aqua (13:30 ascending) satellite since 2002. In general, MODIS measurements employ single-angle viewing, multiple channels (36 channels), high spatial resolution (0.25 to 1.00

km according to channel), and a wide swath (2330 km) enabling daily global coverage for short-wave channels. The MODIS Dark-Target (DT) aerosol retrieval algorithm uses the broader-bandwidth MODIS channels (>20 nm) in the visible to SWIR range. The DT algorithm assumes that land-surface reflectance in the visible range has a linear relationship with SWIR (2.1 μm) surface reflectance, where the atmospheric signal is low (Kaufman et al., 1997), that varies according to the Normalized Difference Vegetation Index (NDVI). This approach is applied to dark land surfaces; e.g., vegetated areas. Ocean-surface

reflectance is based on Fresnel reflectance with the Cox–Munk assumption. The MODIS DT algorithm uses NCEP wind-speed analysis data as input and calculates ocean-surface reflectance according to geometry and wind speed. The MODIS DT AOD at 550 nm is provided at 10 km × 10 km and 3 km × 3 km spatial resolution at nadir after pixel aggregation at the spatial resolution of the reflectance data. In this study, the latest version "Collection 6.1 (C6.1)" data of only the best quality ("Quality Assurance Flag 3") land and ocean 550 nm AOD for both resolutions are used (Gupta et al., 2016; Levy et al.,

2013; Munchak et al., 2013).

## 2.4 MODIS Deep-Blue aerosol product

The MODIS Deep-Blue (DB) aerosol algorithm uses ocean color channels and IR channels to retrieve aerosol optical properties over bright land surfaces. Using the enhanced DB algorithm, MODIS DB AOD is retrieved over arid/semi-arid surfaces, natural vegetation areas, and urban, built-up, and transitional regions using several surface-reflectance calculations. These

calculations use a pre-calculated surface reflectance database with the minimum reflectance technique, a DT-like approach, and a hybrid method over arid/semi-arid surfaces, vegetation, and urban, built-up, and transition surfaces. The MODIS DB algorithm calculates AOD for each of the original L1B 1 km pixels, and aggregates and averages retrieved AOD pixels to 10

km × 10 km resolution at nadir after appropriate masking procedures that differ from those in the MODIS DT and GOCI YAER algorithms. The latest version C6.1 MODIS DB land 550 nm AOD of only the best quality is also used in this study (Hsu et al., 2013; Sayer et al., 2013; Sayer et al., 2019).

## 2.5 MODIS MAIAC aerosol product

The MODIS Multiangle implementation of atmospheric correction (MAIAC) aerosol algorithm performs aerosol retrievals and atmospheric correction over both dark vegetated surfaces and bright deserts land surfaces (Lyapustin et al., 2011a; Lyapustin et al., 2011b). Compared to each scene and pixel-based approach of MODIS DT and DB algorithm, the MAIAC algorithm has a time series analysis and image-based processing. Maximum 16-day data which have multi viewing angle are used to obtain surface bidirectional reflectance distribution function (BRDF) characteristics providing three parameters of the Ross-thick Li-sparse BRDF model. Recent MODIS Collection 6 MAIAC aerosol algorithm was improved in terms of higher spatial resolution of surface characterization from 25 km to 1 km, cloud detection, aerosol model, optimization of LUT-based radiative transfer calculation, and others (Lyapustin et al., 2018). Also, over-water process based on Fresnel reflectance model with the Cox–Munk assumption was added to provide ocean AOD. The MAIAC algorithm uses eight different aerosol models and the same channels with the MODIS DT algorithm for AOD inversion. The latest version C6 MODIS MAIAC 550 nm AOD pixels with "Best Quality" are used in this study.

## 2.6 VIIRS EPS aerosol product

The VIIRS is a sensor onboard the Suomi-NPP satellite, which was launched in October 2011. The general characteristics of VIIRS are similar to those of MODIS, and include single-angle viewing, multiple channels (22 channels), high spatial resolution (375–750 m), and a wide swath (3,040 km) that results in no gaps between adjacent swaths near the equator. Recent VIIRS aerosol products provided by NOAA was updated from the previous Environmental Data Record (EDR) and the Intermediate Product (IP) to the new Enterprise Processing System (EPS) product. The previous VIIRS EDR and IP aerosol retrieval algorithm was similar to the DT algorithm in terms of the coupling of land-surface reflectance in the visible range using the SWIR channel (2.25 μm), ocean-surface reflectance that considers wind speed and direction using the Cox–Munk model with Fresnel reflectance, and a combination of fine- and coarse-aerosol models (Jackson et al., 2013 and Huang et al., 2016). The new VIIRS EPS aerosol algorithm is able to retrieve AOD over both dark and bright as using surface reflectance ratio method (Zhang et al., 2016). The global surface reflectance ratio was obtained as 0.1° × 0.1° spatial resolution using 2 years of VIIRS TOA reflectance. The EPS aerosol product is calculated at a 0.75 km × 0.75 km spatial resolution at nadir. In this study, the VIIRS EPS 550 nm AOD is used.

## 2.7 MISR aerosol product

The MISR is one of the sensors onboard the Terra satellite along with MODIS. Unique characteristics of MISR include multi-channel (four wavelengths, centered at 446, 558, 672, and 866 nm) and multi-angle measurements (nine cameras; nadir, ±26.1°,

±45.6°, ±60.0°, and ±70.5°), which enable better detection of aerosol particle shapes and a distinction between atmospheric and surface signals through calculation of surface bidirectional reflectance factors (BRF). The MISR spatial resolutions at nadir and off-nadir are 250 m × 250 m and 275 m × 275 m, respectively, and the operational MISR aerosol retrieval algorithm provides 550 nm AOD and other optical properties at 17.6 km × 17.6 km in version 22 and 4.4 km × 4.4 km in version 23 after pixel masking and aggregation at the spatial resolution of the reflectance data. Targeted surface conditions for aerosol retrieval are dark ocean, dark vegetation, and bright arid land surfaces. Total 74 aerosol models are ensembled to retrieved AOD in inversion procedure and uncertainty is obtained together. One advantage of MISR measurements is the absence of non-retrieval areas caused by sun-glint effects that are present in nadir-only viewing measurements, such as MODIS. The swath is ~380 km, which is narrower than MODIS, and results in global coverage every 9 days with repeat coverage between 2 and 9 days depending on latitude (2–3 days near the Korean Peninsula). In this study, version 23 AOD at 550 nm is used (Garay et al., 2017; Witek et al., 2018).

## 2.8 AERONET measurements during the KORUS-AQ campaign

To evaluate the various satellite AOD products during the 2016 KORUS-AQ campaign (1 May to 12 June 2016), extensive data from ground-based remote-sensing AERONET sun-sky radiometers were collected from total 33 sites over East Asia including 19 Korean sites (Holben et al., 1998; Holben et al., 2018). Detailed site information, including locations, is available at the AERONET homepage (https://aeronet.gsfc.nasa.gov/new_web/DRAGON-KORUS-AQ_2016.html). AERONET provides high accuracy measurement of AOD with uncertainty of ~0.01 in the mid-visible (Eck et al., 1999). The AERONET Version 3 Level 2.0 AOD at 550 nm all-points data at a few-minutes temporal resolution are used (Eck et al., 2018; Giles et al., 2018). To compare satellite and ground-based AERONET AOD, spatiotemporal collocation is implemented. This study follows the general collocation criteria of Sayer et al. (2014): satellite pixels within a 25 km radius of each AERONET site are spatially averaged, and AERONET data within a ±30 min window around the satellite measurements are temporally averaged. Note that the 10 min interval AHI AOD data are collocated with AERONET AOD within a ±5 min temporal window. Because there are only a few AERONET sites surrounded by ocean, AERONET sites located on a coast are used to validate satellite ocean AOD. Also note that a collocated sample is included in the average if at least one measurement is available.

## 2.9 SONET measurements during the KORUS-AQ campaign

The Sun-Sky Radiometer Observation Network (SONET) operated by the Institute of Remote Sensing and Digital Earth, Chinese Academy of Sciences also provides aerosol optical and microphysical data from ground-based CIMEL sun-sky radiometer measurement using their own retrieval algorithm (Li et al., 2018). Total 5 SONET sites data ("Harbin", "Hefei", "Nanjing", "Shanghai", and "Zhoushan") are used to evaluate satellite land AOD products. The SONET aerosol data and site information during the campaign are available from the AERONET homepage.

## 3 Validation results

### 3.1 Statistical metrics

The statistical metrics as used by Sayer et al. (2014) were also applied here for comparison of satellite AOD measurements over land and ocean using AERONET and SONET and are summarized in Tables 3, 4 and 5. Because the distribution of AOD is non-Gaussian and skewed towards low values, AOD evaluation is difficult using simple statistical techniques. Thus, the metrics applied here consist of the number of matched/collocated data points ($N$), Pearson's linear correlation coefficient ($R$), the root mean square error ($RMSE$), the mean bias ($MB$) error, and the fraction within the expected error of MODIS Collection 5 DT land AOD [$f$ within $EE_{DT}$ or $f$; $EE_{DT} = \pm (0.05 + 0.15 \times$ AERONET AOD)], as suggested by Levy et al. (2007). The range of $EE_{DT}$ consists of upper and bottom linear lines and becomes wider as AOD increases, which reflects increasing AOD uncertainties with AOD in general. Ideally, the ratio within EE corresponds to 1-sigma of the Gaussian distribution data (~ 68 %). The range was obtained from the global evaluation of MODIS DT collection 5 AOD products. Each AOD product has its own expected error envelop as summarized in Choi et al. (2018), but the $EE_{DT}$ was selected to compare different product performance.

### 3.2 Validation of GOCI AOD without and with additional AHI IR cloud masking

The results of land AOD shows increased $R$ from 0.88 to 0.91 and decreased $RMSE$ from 0.17 to 0.15, which indicate improvement due to the removal of overestimation points in high AOD of GOCI from the scene analysis. However, underestimation in low AOD due to surface reflectance uncertainty increases negative $MB$ from –0.04 to –0.07 and constant $f$ of 0.48. The ocean AOD which have smaller uncertainty of surface reflectance shows improvement in most statistical metrics, increasing $R$ from 0.86 to 0.88, decreasing $RMSE$ from 0.14 to 0.12, constant $MB$ of 0.03, and increasing $f$ from 0.66 to 0.68. The GOCI-II planned to be launched in 2020 also doesn't have any IR channels, thus it is essential to adopt additional cloud masking from IR channels of AHI or Advanced Meteorological Imager (AMI) for GOCI-I and GOCI-II aerosol retrieval, to reduce shallow or cirrus clouds contamination. Hereafter, the GOCI V2 YAER AOD with additional IR cloud masking is used as GOCI AOD.

### 3.3 Land AOD validation using AERONET and SONET

Each satellite measures the area within its swath at different times during daylight hours, as listed in Table 1. In contrast to the hourly and 10 min interval measurements of GOCI and AHI, respectively, the LEO satellites observe East Asia only once per day. The overpass time for Terra is at 10:30 Local Time [LT; Coordinated Universal Time (UTC) + 9] and those for Aqua and Suomi-NPP are at 13:30 LT and 13:25 LT, respectively. When measurement times are similar, $N$ can be determined by swath, spatial resolution, and the quality assurance flag, among other factors. Because of gaps arising from its narrow swath, MISR does not fully cover East Asia in one day. This results in the lowest $N$ as 114 for land AOD among LEO sensors. In contrast, the MODIS and VIIRS have wider swaths that cover most of East Asia in one day, resulting in higher $N$ between 800 and

1,348 over land. Not surprisingly GOCI and AHI shows higher $N$ than all LEO sensors due to 1 hour and 10-min temporal resolution respectively.

All land AOD products show high $R$ from 0.87 to 0.93. $RMSE$ range is from lowest 0.12 (MISR) to highest 0.22 and 0.23 (MODIS DT 10 km and 3 km respectively). MODIS DT land AODs also show the largest absolute $MB$ (0.11 and 0.15

respectively). In Fig. 3a, the both MODIS DT 10 km and 3 km products, AHI ESR, and VIIRS AODs are positively biased under low-AOD conditions. In high AOD condition, AHI ESR doesn't show significant bias but MODIS DT and VIIRS still show increased positive biases. The MODIS DT, AHI ESR, and VIIRS algorithms assume a surface reflectance based on an empirical linear relationship between the visible and SWIR channels. Some studies indicate that MODIS Collection 6 DT AOD does not have a noticeable positive $MB$ (Levy et al., 2013; Sayer et al., 2014) globally or over East Asia, but other studies

have reported a positive bias in MODIS C6 DT AOD over East Asia (Choi et al., 2018; Xiao et al., 2016), particularly over urban areas. Although the algorithm was modified to improve AOD accuracy over urban areas starting with Collection 6.1, MODIS DT still overestimates AOD compared with AERONET over East Asia (Gupta et al., 2016). In addition, the DT algorithm is designed for global retrievals and is not optimized for East Asia, which may explain the observed bias. Although the recent VIIRS algorithm calculated regional surface reflectance ratio between visible and SWIR as 0.1° × 0.1° resolution

from globally constant value to improve accuracy, the AOD pixel resolution is not degraded from 0.75 km Level 1B radiance resolution compared to other algorithms because they aim to provide highest resolution aerosol products for air quality applications. Therefore, additional filtering method using spatial variability test or resolution degradation to increase aerosol signal than noise such as cloud is not available. Also, the obtained 0.1° × 0.1° surface reflectance ratio database could still miss a smaller scale urban surface heterogeneity.

In contrast, GOCI and AHI MRM AOD shows negative $MB$ of –0.07 and –0.06 and The GOCI and AHI MRM aerosol retrieval algorithms obtain surface reflectance using the minimum reflectivity technique with monthly samples of Rayleigh-corrected reflectance (RCR). Although this technique is designed to obtain cloud-free and aerosol-free conditions by finding dark pixels within the composite dataset, the calculations can still be affected by aerosols and clouds, resulting in overestimated surface reflectance. The climatological surface-reflectance database of GOCI did not show significantly negative biased AOD between

2011-2015 according to the validation study of Choi et al. (2018). This negative bias in 2016 may be due to a sensor calibration issue or degradation, but the exact cause is difficult to diagnose and remains unknown.

MISR and MAIAC land AOD shows highest $R$ as 0.93, small $RMSE$ as 0.12 and 0.15, small MB as –0.02 and 0.05, which result in highest $f$ as 0.81 and 0.68, respectively. A common characteristic between MISR and MAIAC is a multi-angle measurement capability (9 cameras of MISR and 16 days composite of MAIAC) which enables to distinguish surface and

aerosol signals well. The MODIS DB used a hybrid method of surface reflectance ratio between visible and SWIR and pre-calculated surface reflectance using minimum reflectivity technique, thus the bias is not large as DT or GOCI. These DB and MISR results are consistent with previous studies (Choi et al., 2018; Garay et al., 2017; Sayer et al., 2014).

Because of limited number of SONET sites, the $N$ is about 5-7 % of AERONET. Statistical metrics using SONET tend to be similar to the comparison with AERONET, but the values of metrics show worse agreement from SONET than AERONET

especially in terms of *RMSE* and *f*, which is similar to the result of Choi et al. (2018). The AHI ESR, MODIS DT, and VIIRS show consistently positive *MB* from 0.08 to 0.15. GOCI shows lower *R* of 0.75, higher *RMSE* of 0.22, worse *f* of 0.29, but better *MB* of –0.03, compared to the results with AERONET. It can be attributed to high variation of GOCI climatological surface reflectance uncertainty according to sites. MAIAC shows lowest *RMSE* of 0.15 and *MB* of 0.03 and highest *f* of 0.67.

MISR also show high *f* of 0.60 despite a small *N* of 5. DB shows slightly lower *R* of 0.82, higher *RMSE* of 0.21, similar *MB* of 0.08, and lower f of 0.46 compared to the results using AERONET. The aerosol retrieval algorithm, maintenance, calibration of SONET is different from those of AERONET, thus it is difficult to explain the difference between the two results using AERONET and SONET. Chinese sites seem to have more difficulties to retrieve aerosol properties from most satellite instruments.

## 3.4 Ocean AOD validation using AERONET

The target area for ocean aerosol retrievals differs among the various algorithms. The MODIS DT, MISR, MAIAC, and VIIRS algorithms retrieve aerosol properties only for dark-ocean pixels, which means that surface pixels that are not completely dark, such as those containing shallow or turbid water, are masked. The GOCI/AHI Yonsei aerosol algorithms are also designed to

retrieve aerosols over dark pixels, but they include moderately turbid water pixels by considering the climatological ocean-surface reflectance based on minimum-reflectance techniques in the GOCI and AHI MRM algorithms and by considering chlorophyll-*a* concentrations in the AHI ESR algorithm. Because the ocean AOD validation was conducted using coastal AERONET AOD observations, *N* is higher for GOCI and AHI (230–237) than for the LEO ocean AOD observations (13–111), with the exception of the VIIRS ocean AOD (252). Sun-glint areas also likely contribute to the difference in *N*. Single-

angle viewing LEO satellite measurements, such as MODIS, exclude bright ocean-surface pixels because of sun-glint close to nadir, where most pixels are screened out, as is evident in Fig. 3. This occurs daily near the Korean Peninsula and results in most transported aerosol plumes around Korea not being measured with continuous spatial coverage. Although the VIIRS is also a single-angle viewing instrument, its broader swath results in more ocean pixels being retrieved than is the case for MODIS. The MISR instrument minimizes sun-glint effects over ocean pixels through multi-angle viewing, but still has low *N*

because of its narrow swath. The sun-glint areas of the GEO satellites are located near the equator, have a circular shape, and shift from east at sunrise to west at sunset. Most SE Asian countries, including the Philippines, Malaysia, and Thailand, are affected by this sun-glint screening in ocean AOD from GEO satellites, whereas most northeast (NE) Asian countries, including China, Korea, and Japan, are unaffected. Thus, GOCI and AHI provide spatiotemporally continuous aerosol measurements across land and ocean over NE Asia where dense aerosol plumes of varying composition are transported from mainland Asia

to the Pacific.

According to most validation metrics, ocean AOD products are more accurate than those over land. This difference leads to generally lower errors for ocean AOD compared to their respective over land retrievals, based on AERONET AOD measurements (Fig. 3b). The sign of the ocean AOD error in the low-AOD range is the same as that of the land AOD error for

all products; i.e., negative in GOCI and AHI MRM, and positive in DT and AHI ESR. The MAIAC, VIIRS, GOCI, and AHI products have high accuracy, as evidenced by a low *RMSE* (0.12–0.13) and a near-zero *MB* (−0.03 to 0.04), resulting in a high *f* (0.612–0.769). The MODIS DT 3 km ocean AOD product has a larger positive *MB* (0.10) than the 10 km product (0.06) similar to MODIS DT land AOD.

In summary, most LEO and GEO aerosol products over East Asia are highly accurate based on a comparison with AERONET with high *R* (0.84–0.93) and low *RMSE* (0.12–0.17), but have unique bias patterns related to the surface-reflectance assumptions in each algorithm. This leads to positive biases for MODIS DT and AHI ESR AOD, negative biases for GOCI and AHI MRM AOD, and small biases for the other products. The coverage also differs between single-angle and multi-angle measurements, and with swath size and orbit types, resulting in a range of *N* values.

**4 Transport events during the campaign**

**4.1 Analysis of the period 18–21 May 2016 over Hokkaido, Japan**

Noticeable aerosol transport was observed over Hokkaido, Japan, during the period 18–21 May 2016. Although GOCI and AHI AODs were retrieved at 1 h and 10 min temporal resolutions, respectively, only data for 09:30 and 13:30 LT are presented in Fig. 4 for comparison with MODIS, MISR, and VIIRS distributions. A time series of satellite AODs collocated with

AERONET AOD from the Hokkaido University, which is located at 142.34°E longitude, 43.08°N latitude, and 43.08 m altitude above sea level, is presented in Fig. 5.

As the dense smoke aerosol plume (AOD > 2.0 at the center) generated from the Russian forest fires was transported to Hokkaido continuously from morning to afternoon on 18 May, AERONET AOD at Hokkaido University increased rapidly from 0.1 to 1.4, and the GOCI and AHI successfully detected this abrupt increase. The MODIS and VIIRS instruments also

detected increasing AOD accurately but the first and last AODs during the day were 0.6 and 1.1 at 10:30 and 13:30 LT, respectively, and therefore did not capture the full diurnal variation detected by AERONET, GOCI, and AHI. The increase of AOD at Hokkaido on 18 May was anticipated from the southward movement of the plume revealed by the GOCI and AHI measurements. On 19 May, the plume remained over Hokkaido and the spatial distribution changed little during daylight hours. The AOD observed by AERONET decreased from 1.3 to 0.9, and the GOCI and AHI instruments detected this change, but

with a slight overestimation during the morning. The VIIRS, MODIS DT and DB AODs are higher as about 1.5 and the MISR AOD is lower as 0.9 than the AERONET value. On 20 May, the AERONET AOD began to increase to 1.0 at 06:00 LT, peaked up to 1.3 at 12:00–13:00 LT, and sharply decreased down to ~0.6 at 18:00 LT. The GOCI and AHI retrievals again followed this variation well, beginning at 09:00 LT. The AHI also detected the AOD peak well, but the MODIS DT and DB overestimated AOD compared with AERONET. The pixels involved did not include cloud edges, so this difference in AOD

was not due to cloud contamination. AE between 440 nm and 870 nm at Hokkaido university AERONET site was around 1.95, and SSA at 440 nm was about 0.9, which means that those aerosols were small particle size and less absorbing as aging smoke plume. On 21 May, the dense AOD plume was bifurcated into two: one moved out to Pacific Ocean and the other to the south–

west direction of Hokkaido. The AERONET AOD over the Hokkaido University site decreased slightly from 0.6 to 0.4 and most products detected these low-AOD conditions well.

It is very hard to figure out exact reason of overestimation of MODIS DT, DB, VIIRS, and MAIAC AOD over this plume despite reasonable accuracy from AERONET validation. The statistical metrics of MODIS DT, DB, MAIAC, and VIIRS

validation at Hokkaido University site during the campaign show very high $R$ (0.96-0.98) and small offset of linear regression equation (–0.03 to 0.03) but higher slope than one (1.22–1.43) reveling high $MB$ (0.12–0.18). Small offset of linear regression equation represents lower surface reflectance error in AOD validation. With this condition, higher slope generally means that AOD overestimation due to the aerosol model assumption generally (Hyer et al., 2011) if cloud masking is working well. This transport case results in also high AOD, where uncertainty of aerosol model can be emphasized. Therefore, possible reason of

overestimation is due to an aerosol model assumption such as microphysical properties.

It can be summarized that overall evaluation is not matched with individual site or case over East Asia because of complexity of surface condition and dynamic aerosol types. Additionally, MODIS and VIIRS do not provide spatially continuous AOD distributions because of sun-glint masking over ocean areas near Hokkaido, making identification of plume sources and transport pattern difficult. In contrast, GEO can avoid sun-glint area over mid-latitude area. Sun-glint is a bright ocean surface

due to the reflected solar radiance, which is brighter in nadir viewing angle. Due to the measurement geometry, single-angle viewing LEO sensors such as MODIS and VIIRS have the sun-glint pixels in the middle of swath generally. In contrast, GEO has the sun-glint pixels as a circle shape centered at equator because GEO sensors are located at the equator. Because of multi-temporal measurement without sun-glint pixels, GEO such as GOCI and AHI can detect these transported aerosol plumes across ocean with more continuous spatiotemporal distribution than LEO.

**4.2 Analysis of 25 May 2016 and 5 June 2016 cases over Yellow Sea and Korean Peninsula**

Next, two heavy aerosol loading cases over Korean Peninsula are analyzed as in Fig. 6. During the campaign, the first noticeable increase in PM above the Korean national air-quality standard (50 μg m$^{-3}$ before April 2018; now 35 μg m$^{-3}$) occurred on 25 May 2016 and resulted in dense aerosol conditions around the Korean Peninsula. In the morning, high AOD values ranging from 0.8 to 2.0 were measured by GOCI, AHI, and MISR over the Yellow Sea located to the west of the

Korean Peninsula. A few land pixels in the southwestern Korean Peninsula adjacent to this dense aerosol plume also showed high AOD values of ~1.0. Land pixels in the northwestern Korean Peninsula and adjacent ocean pixels were screened out because of clouds. Very low AOD values (0.0–0.3) were observed at other land pixels over the eastern Korean Peninsula had. Most ocean AOD pixels are screened out from MODIS DT and MODIS MAIAC because of sun glint. MISR detect the plume over the Yellow Sea, not affected by sun glint with multi-angle imaging capability. As the plume continuously moved

eastward, high AOD plume entered over land pixels in the Korean Peninsula and a steep zonal gradient of AOD was evident near 127°E in the afternoon. To evaluate the temporal AOD transportation quantitatively, Hovmöller diagram of GOCI and AHI MRM AOD pixels within a box area (123°E–128°E and 35°N–38°N) were constructed by averaging meridionally at a 0.02° longitude interval as shown in Fig. 7a and c. The peak at 09:30 LST was located at ~123.5°E and moved continuously

eastward to 123.8°E, 124.4°E, 124.8°E, 125.0°E, 125.5°E, 125.8°E, and 126.3°E at 1 h intervals until 16:30 LST. This transport corresponds to easterly zonal wind direction at 850 hPa of the Fifth generation of European Centre for Medium-Range Weather Forecasts (ECMWF) atmospheric reanalyses of the global climate (ERA5; Copernicus Climate Change Service, 2017). The AOD over the Yellow Sea (123°E–126°E) decreased from 1.2 to 0.9 as the plume passed over. In

contrast, the AOD over the Korean Peninsula (126°E–129.5°E) increased gradually, particularly over 127°E in the western Korean Peninsula where it increased from 0.3 to 0.8. The eastern Korean Peninsula (128°E–129.5°E) was not affected by the plume during daylight hours, and the AOD remained low (0.2–0.3). More detailed feature can be found from higher temporal resolution of AHI than GOCI. Transport speed of plume center (AOD > 1.1) can be calculated as about 39 km/h (e.g. 10.9 m/s) from 123.5°E to 126.0°E during 09:00–15:00 LT, which is similar to the wind speed at 850hPa in Fig 7a and c.

Compared with conditions on 25 May 2016, the overall AODs on 5 June 2016 over the Yellow Sea and Korean Peninsula was low (0.1–0.2) and the AOD over the Seoul Metropolitan Area (SMA) near 127°E and 37°N was about 0.4-0.6 from GOCI and AHI MRM in the morning as in Fig. 6. The focus here is on SMA AOD, which increased up to 1.0 and dispersed out to surrounding areas in the afternoon. The quantities of MISR AOD in the morning (around 0.4-0.5) and VIIRS AOD in the afternoon (around 1.0) over SMA area is analogous with GOCI, AHI MRM, and MODIS MAIAC. In contrast, changes

in AOD was less significant from AHI ESR, MODIS DT, and MODIS DB, because morning AODs were higher (around 1.0) than others (around 0.4-0.5). Because the periphery of the SMA remained under low-AOD conditions and aerosol transport from China through the Yellow Sea was not detected, this increase can be attributed to local emissions. A distinct pattern is evident in the temporal changes of meridional mean AODs shown in Fig. 7b and d. The mean AODs in the region 125.5°E–127.0°E gradually increased from 0.2 to 0.5, whereas the AODs in other areas, including the Yellow Sea and

eastern Korean Peninsula, remained constant during daylight hours. Unlike conditions on 25 May, the dense aerosol plume on 5 June grew rapidly over a short period of time from local-area emissions and was transported to the Yellow Sea. The wind was easterly and speed in the afternoon was weaker than the case of 25 May 2016, which resulted in less dispersion pattern of local emission compared to the previous case.

The two events analyzed in this section involved rapid changes in hourly AOD, but have noticeably different spatiotemporal

characteristics, leading to high-AOD conditions that are attributed to either long-range transboundary transport from China or local emissions in Korea (Lee et al., 2019). To accurately assess these types of events, spatiotemporally continuous measurements with minimal data gaps are required, which are currently possible only from GEO measurements.

## 5 Comparison of spatial distribution and daily AOD integration

### 5.1 Averaging daily and campaign-period AOD on a common grid

Because the various satellite AOD products were validated using AERONET, results are only valid for specific ground sites. A comparison between satellite products can provide the relative difference in AOD for each pixel, but a direct comparison between satellite products of Level 2 (L2) data is difficult because they differ in spatial coverage, measurement time, and

spatiotemporal resolution. For this reason, each L2 AOD product was regenerated as a daily average value on the spatial grid of the Level 3 (L3) products. Although some products are available in the L3 format, the methods and criteria used in their L3 calculation differ considerably. Thus, a simple and commonly used method is applied here to generate daily L3 AOD. The spatial domain is set to 110°E–150°E and 20°N–50°N, and the grid resolution is set to 0.5° × 0.5°. For the aggregation,

textural filtering described by Zhang and Reid (2006) and Hyer et al. (2011) is used to reduce random error through averaging. Then, AOD pixels within a grid cell are filtered if the number of retrieved AOD pixels is <3 or the coefficient of variation of AOD within the grid cell is >0.5 and the mean AOD is >0.2. The number of pixels used to calculate one 0.5° × 0.5° pixel is determined by the spatial resolution of the L2 AOD data and the number of filtered pixels. After aggregation of each distribution to the L3 grid, the distributions for each day are averaged to a daily mean value. Temporal resolution and

swath determine the maximum number of temporal samples used in the daily mean value: 8 for GOCI (1 h temporal resolution), 47 for AHI (10 min temporal resolution), 2–4 for MODIS (Aqua and Terra), and 1 or 2 (swath-overlapped pixels) for VIIRS and MISR. Note that the 10 km product is used for MODIS DT instead of 3 km because of lower positive bias.

## 5.2 Comparison of observation frequency during the campaign

The number of L2 AOD pixel samples within each 0.5° × 0.5° grid cell over 110–150°E and 20–50°N during the campaign period is calculated and the maximum number of them is denoted as $N_{max}$. $N_{max}$ is high when the spatial and temporal resolutions are high and when fewer pixels are masked because of the presence of clouds or uncertain surface reflectance. The calculated mean AOD can be more statistically representative when $N_{max}$ is high. Highly uncertain AOD values can be removed during the spatiotemporal averaging steps, but these pixels can still lead to high uncertainty when temporally averaged mean AOD is

calculated from only a few samples. Thus, we can determine the reliability of AOD values for each region and for each product using the number of L2 pixel samples. The $N_{max}$ can be sorted as descending order as MAIAC ($1.3 \times 10^5$), AHI ESR ($7.3 \times 10^4$), AHI MRM ($7.1 \times 10^4$), VIIRS ($6.2 \times 10^4$), GOCI ($1.2 \times 10^4$), MISR (668), MODIS DT 10 km (661), and MODIS DB (556). The high $N_{max}$ of MAIAC and VIIRS is from high spatial resolution of 1 km and 0.75 km respectively, and the high $N_{max}$ of AHI and GOCI is from high temporal resolution of 10 minutes and 1 hour, respectively.

The number of L2 pixels for each grid can be normalized by $N_{max}$ to compare relative sampling frequency (RSF) ranged from 0 to 1 between products as Fig. 8. Most products have more samples over the Korean Peninsula and eastern China than over Japan. Possible reason of lower sampling in Japan is higher amount of cloud due to its adjacency of the Pacific Ocean, or a combination of lower AOD condition and very high mountains and therefore slope effects, plus high spatial variance of AOD with higher values in the valleys and lower over the mountain peaks. The RSF of GOCI is about 0.6–0.7 over the Korean

Peninsula, with some discontinuity between land and ocean. The negative GOCI AOD bias under low-AOD conditions from land-surface reflectance effects results in retrieved AOD values of less than −0.05, which are screened out of the final product. This bias has less of an effect on the land pixels of GOCI. As AHI AOD is not affected by these errors, continuously high RSF over land and ocean surfaces exists in the Korean Peninsula. The RSF of MISR includes a discontinuity between paths that

results in a more discontinuous AOD distribution compared with the other products. The MODIS DT and DB have similar distributions of RSF over land, except over Manchuria (0.6–0.8 of DB but 0.0-0.2 of DT). The MODIS DT has small RSF as 0.0–0.4 over the Yellow Sea because of turbid water and sun-glint masking. The RSF distribution of VIIRS is similar to that of MODIS DT, but is higher over the ocean because of its broad swath thereby avoiding sun glint at times. MAIAC shows

similar pattern of RSF with VIIRS but higher value over bright Manchuria surface as close to 1. Also, all products show low RSF over southern east of China (left bottom land area of each panel) as 0.0–0.2, which can result in higher uncertainty over that area even if all products are integrated.

## 5.4 Daily AOD fusion

Retrieved satellite AOD errors can be classified into two types: random error and bias. Although some algorithms, such as the

optimal estimation method, can provide an estimated random error or uncertainty quantitatively (e.g., Jeong et al., 2016), the random error and bias of retrieved AOD can be assessed only over AERONET sites, making it difficult to quantify and validate uncertainties for all pixels. As errors were found to be distributed equally around zero for land and ocean surfaces during the validation using AERONET data, the combined AOD is calculated by selecting the median value from the daily 0.5°×0.5° gridded mean AOD among different products. The reason of selecting median value among products is based on well

distributed bias patterns from Fig 3 and to exclude extreme values. The mean of daily AOD fusion result is presented in Fig. 9, and there are patterns of high AOD (0.8-1.0) in eastern China, low AOD in Japan (0.2–0.4), high AOD in western Korea (0.5–0.6), and low AOD in eastern Korea (0.2–0.4). The ratio to be selected median value to represent fusion AOD for each daily grid can be calculated per each product. When the area is limited to the smallest GOCI domain, the selection ratio can be sorted as descending order: AHI ESR (27.2%), AHI MRM (25.2%), GOCI (16.2%), MODIS MAIAC (10.8%), VIIRS

(10.7%), MODIS DT (5.7%), MODIS DB (2.8%), and MISR (1.4%). Note that MODIS DB can be underestimated due to the lack of ocean AOD. Because GEO measurements yield more samples that contribute to the daily representative AOD, the AHI and GOCI account for a higher fraction of the combined AOD. Among LEO products, MAIAC and VIIRS having higher spatial resolution with wide swath show higher selection ratio.

Evaluation of the daily average AOD for each product and the combined AOD using daily AERONET AOD is presented in

Fig. 10. The closest grid point to each AERONET site is selected for the comparison. The number of selected grid points is 870 (AHI), 768 (GOCI), 677 (MODIS MAIAC), 658 (VIIRS), 436 (MODIS DT), 303 (MODIS DB), and 106 (MISR). Most products show similar bias patterns to the Level 2 pixel-level validation. For instance, GOCI is negatively biased and MODIS DT is positively biased. However, when we combined these all products to the fusion AOD, it has a higher $f$ (0.61) than the individual products, except for MISR (0.77) and MAIAC (0.65), and a high $N$ (869), low $RMSE$ (0.16), and high $R$ (0.87). This

satisfies our objective to generate a more representative AOD field includes more pixels with high accuracy than do the individual AOD products.

## 5.3 Comparison of difference between each product and fusion AOD product

The validation using AERONET is only available over a few specific grids. Thus, the difference between each product AOD and fusion AOD of Fig. 9 is calculated and compared as Fig. 11. GOCI shows relative lower AOD than fusion AOD over land pixels over Korea Peninsula and Japan about 0.2, and significantly over southern east China up to 0.4 and higher AOD over Manchuria land area about 0.3. AHI MRM and ESR shows least difference overall over most area and its related to the highest selection ration of AHI products for fusion AOD. Interesting feature is that positive-negative pattern is opposite between MRM and ESR over most grids, which is agreed with improved accuracy in AHI when these two products are merged in Lim et al. (2018). Narrow swath of MISR leads to a broad gap between paths, and the discontinuity of the MISR L2 AOD data is noticeable along the swath boundary. More period averaging seems to be required to analyze MISR AOD characteristics compared to others. The difference between VIIRS AOD and fusion AOD is quite similar to that of AHI ESR as higher AOD over southern east China (0.3) and lower AOD over northern east China (0.2). MODIS DT shows quite noise patterns over Yellow Sea compared to others, which can be related to the lower sampling frequency due to sun glint and turbid/shallow water masking with coarse pixel resolution (10 km). MODIS DB shows similar pattern to GOCI over land except for Manchuria. Because Manchuria area is bright surface, MODIS DB, MODIS MAIAC, VIIRS, and MISR can have better accuracy than others. The MAIAC shows generally less difference with fusion AOD except for higher AOD over ocean grids near Chinese coast. Lyapustin et al. (2018) also notes that current masking of MAIAC misses several coastal waters with high sediments where AOD retrievals often show a high bias.

## 6. Discussion and conclusions

In this study, we compare spatiotemporal characteristics of three GEO AOD products (GOCI, AHI MRM, and AHI ESR) and four LEO AOD products (MODIS DT, MODIS DB, MODIS MAIAC, MISR, and VIIRS), and validate each product using the AERONET version 3 and SONET dataset for the 2016 KORUS-AQ campaign. Most AOD products have high accuracy and wide coverage over East Asia, but each has individual unique characteristics (e.g. detailed accuracy and sampling frequency). Although Choi et al. (2018) showed that GOCI AOD is reliably accurate for the period 2011–2015, it is negatively biased during the 2016 campaign period. This difference in accuracy may be attributable to changes in climatological surface reflectance or calibration drift. Improvement of surface reflectance including these calibration drift or surface reflectance change is required. The DT method used in AHI ESR and MODIS DT AOD retrievals results in a positive bias and higher AOD over East Asia compared to other products. The MISR AOD has smaller coverage than MODIS and VIIRS, but the AOD accuracy is higher than for the other products because of an improved surface-reflectance treatment that takes advantage of multi-angle measurements. However, it also seems that the MISR retrievals often screen out the highest AOD events, thereby biasing the sampling in this region. MISR uses neither SWIR channels, nor pre-calculated surface reflectance, the algorithm does not retrieve AODs if aerosol signal is too high to get surface signals consistently. The range of MISR AOD product is set to be from 0.0 to 3.0 according to Witek et al. (2018). The maximum value is lower than others such as 3.6 of GOCI and 5.0

of MODIS. These dynamic range and accuracy differences are due primarily to algorithm design, which is optimized for particular sensor specifications, such as the available channels, and are not related to orbit types. The MAIAC AOD shows high accuracy ($f$ of 0.68 and 0.67 over land AEORNET sites and SONET sites respectively) during the campaign and best spatial coverage among MODIS products.

As GOCI and AHI AOD can be retrieved with high accuracy at near-real-time, the highly variable AOD conditions over East Asia, including transport from Russia to Japan, transport from China to Korea, and local emissions in the SMA area and subsequent transport to the Yellow Sea, can be successfully detected. This results in more representative daily AOD values. A combined AOD using GEO and LEO data is also tested using a median-value selection at the daily scale with a 0.5° × 0.5° grid resolution. The combined AOD has a more spatially continuous distribution and higher accuracy than do the individual

products. Such a combined product reduces bias in aerosol measurements and will be of use in the evaluation of air-quality models.

Although the validation using AERONET data reveal relative characteristics among the various AOD products in terms of accuracy, it is insufficient to thoroughly investigate these characteristics. Each algorithm includes subjective criteria, such as those used in cloud masking, surface-reflectance determination, aerosol model selection, inversion methods, and quality

control. For example, the possible AOD range that can be retrieved and provided as the final AOD product varies among GOCI (−0.05 to 3.6), AHI (−0.05 to 3.5), MODIS DT (−1.0 to 5.0), MODIS DB (0.0 to 5.0), MODIS MAIAC (–0.1 to 5.0) and MISR (0.0 to 3.0). The quality flag is also determined subjectively. This results in differences in the identification of severe pollution events (e.g., AOD > 3.0 or > 5.0) among the various products. The target regions of the GEO and LEO measurements also differ. Aerosol retrieval algorithms for LEO measurements have been developed for global coverage, but those for GEO

measurements only consider the accuracy within specific regions. Because the validation datasets differ, algorithm improvement proceeds differently among the various algorithms. Thus, the integration of multiple-AOD products requires a comprehensive understanding of each product in the set. To reduce uncertainties arising from the use of different algorithms, the same algorithm can be applied to several sensors, just as the DB algorithm is applied to AVHRR, SeaWiFS, MODIS, and VIIRS measurements (Hsu et al., 2013; Lee et al., 2015; Sayer et al., 2012; Sayer et al., 2017), and the DT algorithm is applied

to MODIS, VIIRS, and planned to be applied to AHI and ABI. Additionally, LEO and GEO aerosol measurements can be integrated at the resolution of radiance data as Level 1B, not retrieved AOD products, as a concept of the multi-angle measurement. This integration will enable the retrieval of other aerosol optical properties, such as particle shape or absorptivity, which can be used to evaluate aerosol optical effects along with chemical composition.

This study focuses only on the spring season of 2016 when the KORUS-AQ campaign was conducted. An extended long-term

study will be required to evaluate monthly or seasonal mean AOD trends of GEO and LEO measurements and combined AOD products. Additionally, the integration of multiple datasets may be improved by a consideration of pixel-level uncertainties and varying error characteristics, pixel size, and pixel shape, and the application of more advanced statistical techniques. Other optical properties, such as the Ångstrom exponent and single scattering albedo, should also be investigated along with AOD in future studies.

**Author contributions**

MC, HLim, SL, JK, TE, BH, MG, EH, PS, HLiu

MC and JK designed the data analysis. MC, HLim, SL, JK, EH, PS carried out the GOCI and AHI data production, distribution, and analysis. MC, HLim, SL, JK, TE, and BH carried out an installation, maintenance, and data analysis of the AERONET measurement during the 2016 KORUS-AQ campaign. MG provided the MISR AOD data and contributed to the data analysis. HLiu provided the VIIRS AOD data and contributed to the data analysis. MC and JK wrote the manuscript with comments from all coauthors.

**Data availability**

The GOCI and AHI Yonsei aerosol retrieval data during the KORUS-AQ campaign are available from https://www-air.larc.nasa.gov/missions/korus-aq/ (last access: 07 July 2019). More period of the GOCI and AHI Yonsei aerosol products are available through personal communication with the corresponding authors of the present paper. The AERONET and SONET data during the KORUS-AQ campaign are available from https://aeronet.gsfc.nasa.gov (last access: 07 July 2019). The MODIS Dark-Target, Deep-Blue, and MAIAC aerosol data and MISR aerosol data can be available from https://earthdata.nasa.gov/ (last access: 07 July 2019). The VIIRS EPS aerosol data can be available from one of authors of the present paper, Hongqing Liu (hongqing.liu@noaa.gov), and will be available from https://www.avl.class.noaa.gov/ . The ECMWF atmospheric reanalyses data are available from https://cds.climate.copernicus.eu (last access: 07 July 2019).

**Competing interests**

The authors declare that they have no conflicts of interest.

**Acknowledgements**

We thank all principal investigators and their staff for establishing and maintaining the AERONET and SONET sites used in this investigation. We also thank the MODIS, MISR, and VIIRS science teams for providing valuable data for this research. This research was supported by the National Strategic Project–Fine Particle of the National Research Foundation of Korea (NRF) funded by the Ministry of Science and ICT (MSIT), the Ministry of Environment (ME), and the Ministry of Health and Welfare (MOHW; NRF-2017M3D8A1092021). Some research tasks were supported by the NASA ROSES-2013 Atmospheric Composition: Aura Science Team program and NASA Headquarter Directed Research and Technology Development Task (grant number: NNN13D455T, manager: Kenneth W. Jucks and Richard S. Eckman). A portion of this research was carried out at the Jet Propulsion Laboratory, California Institute of Technology, under a contract with the National Aeronautics and

Space Administration. The editor and two anonymous reviewers are thanked for numerous useful comments, which improved the content and clarity of the manuscript.

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

**Tables**

**Table 1. Characteristics of multi-sensor aerosol products.**

| Sensor / Platform (orbit type) | GOCI / COMS (GEO) | AHI / Himawari-8 (GEO) | MODIS / Terra, Aqua (LEO) | VIIRS / Suomi-NPP (LEO) | MISR / Terra (LEO) |
|---|---|---|---|---|---|
| Swath for LEO or local coverage for GEO | 2500 km × 2500 km area over East Asia centered at 130°E, 36°N | Full disk centered at 140.7°E | 2330 km | 3040 km | 380 km |
| Algorithm version | Yonsei Aerosol Retrieval version 2 | Yonsei Aerosol Retrieval | Dark-Target Collection 6.1; Deep-Blue Collection 6.1 (land-only); Multi-Angle Implementation of Atmospheric Correction (MAIAC) Collection 6 | Enterprise Processing System (EPS) | Version 23 |
| Measurement time (Local Standard Time) | 1 h interval from 09:30 to 16:30 (totally 8 times during daylight) | 10 min interval for full-disk measurements (only 09:00–16:50 in this study) | 10:30 for Terra 13:30 for Aqua | 13:25 | 10:30 |
| Spatial resolution of aerosol products (nadir point for LEO) | 6 km × 6 km | 6 km × 6 km | 10 km × 10 km and 3 km × 3 km for DT; 10 km × 10 km for DB; 1 km × 1 km for MAIAC | 0.75 km × 0.75 km | 4.4 km × 4.4 km |
| References | Choi et al. (2018); Choi et al. (2016) | Lim et al. (2018) | Gupta et al. (2016); Hsu et al., 2013; Levy et al. (2018); Lyapustin et al., 2018; Sayer et al., 2013 | Huang et al. (2016); Jackson et al. (2013); Zhang et al., 2016 | Garay et al. (2017); Witek et al. (2018) |

**Table 2. IR cloud masking processes of AHI YAER algorithm. Note that "High latitude" in step 1 corresponds to the 1st, 2nd, 9th, and 10th segments from the North of the AHI observation segments (total 10 segments for the full disk area), and "Mid-low latitude" in step 1 corresponds to the other 6 segments.**

| Steps | Brightness temperature (BT) difference (BTD) test conditions | Classifications |
|---|---|---|
| 1 | BTD between 11.2 and 12.4 μm | Cloud |
| | • Land: BT (11.2 μm) – BT (12.4 μm) < 1.5 K | |
| | • High latitude ocean: BT (11.2 μm) – BT (12.4 μm) < –1.0 K | |
| | • Mid-low latitude ocean: BT (11.2 μm) – BT (12.4 μm) < 0.5 K | |
| 2 | BTD between 12.4 and 13.3 μm | High level cloud |
| | • Land and ocean: BT (12.4 μm) – BT (13.3 μm) < 11 K | |
| 3 | BTD between 8.6 and 6.9 μm | Low level cloud |
| | • Land and ocean: BT (8.6 μm) – BT (6.9 μm) < –10 K | |
| 4 | BTD between 11.2 and 8.6 μm | Cirrus cloud |
| | • Land and ocean: BT (11.2 μm) – BT (8.6 μm) < 0 K | |

**Table 3. Validation statistics for land AOD products using AERONET.**

| Products (resolution) | $N$ | $R$ | $RMSE$ | $MB$ | $f$ within $EE_{DT}$ |
|---|---|---|---|---|---|
| GOCI (6 km) | 4,292 | 0.91 | 0.15 | −0.07 | 0.48 |
| AHI MRM (6 km) | 19,160 | 0.91 | 0.14 | −0.06 | 0.58 |
| AHI ESR (6 km) | 19,174 | 0.90 | 0.16 | 0.07 | 0.54 |
| MODIS DT (10 km) | 988 | 0.87 | 0.22 | 0.11 | 0.51 |
| MODIS DT (3 km) | 1,312 | 0.88 | 0.23 | 0.15 | 0.43 |
| MODIS DB (10 km) | 851 | 0.88 | 0.17 | 0.07 | 0.53 |
| MODIS MAIAC (1 km) | 1,348 | 0.93 | 0.15 | 0.05 | 0.68 |
| MISR (4.4 km) | 114 | 0.93 | 0.12 | −0.02 | 0.81 |
| VIIRS (0.75 km) | 800 | 0.87 | 0.16 | 0.07 | 0.59 |

**Table 4. Validation statistics for land AOD products using SONET.**

| Products (resolution) | $N$ | $R$ | $RMSE$ | $MB$ | $f$ within $EE_{DT}$ |
|---|---|---|---|---|---|
| GOCI (6 km) | 287 | 0.75 | 0.22 | −0.03 | 0.29 |
| AHI MRM (6 km) | 922 | 0.85 | 0.17 | −0.03 | 0.43 |
| AHI ESR (6 km) | 926 | 0.89 | 0.18 | 0.09 | 0.58 |
| MODIS DT (10 km) | 50 | 0.91 | 0.21 | 0.08 | 0.48 |
| MODIS DT (3 km) | 83 | 0.87 | 0.25 | 0.15 | 0.37 |
| MODIS DB (10 km) | 59 | 0.82 | 0.21 | 0.08 | 0.46 |
| MODIS MAIAC (1 km) | 89 | 0.88 | 0.15 | 0.03 | 0.67 |
| MISR (4.4 km) | 5 | 0.99 | 0.18 | −0.09 | 0.60 |
| VIIRS (0.75 km) | 58 | 0.90 | 0.23 | 0.11 | 0.43 |

**Table 5. Validation statistics for ocean AOD products using AERONET.**

| Products (resolution) | $N$ | $R$ | $RMSE$ | $MB$ | $f$ within $EE_{DT}$ |
|---|---|---|---|---|---|
| GOCI (6 km) | 1,766 | 0.88 | 0.12 | 0.03 | 0.68 |
| AHI MRM (6 km) | 7,575 | 0.87 | 0.13 | −0.03 | 0.64 |
| AHI ESR (6 km) | 7,663 | 0.84 | 0.13 | 0.02 | 0.76 |
| MODIS DT (10 km) | 85 | 0.91 | 0.12 | 0.06 | 0.61 |
| MODIS DT (3 km) | 205 | 0.87 | 0.16 | 0.10 | 0.49 |
| MODIS MAIAC (1 km) | 248 | 0.88 | 0.10 | 0.00 | 0.72 |
| MISR (4.4 km) | 13 | 0.63 | 0.15 | 0.04 | 0.77 |
| VIIRS (0.75 km) | 252 | 0.84 | 0.12 | 0.04 | 0.69 |

**Figures**

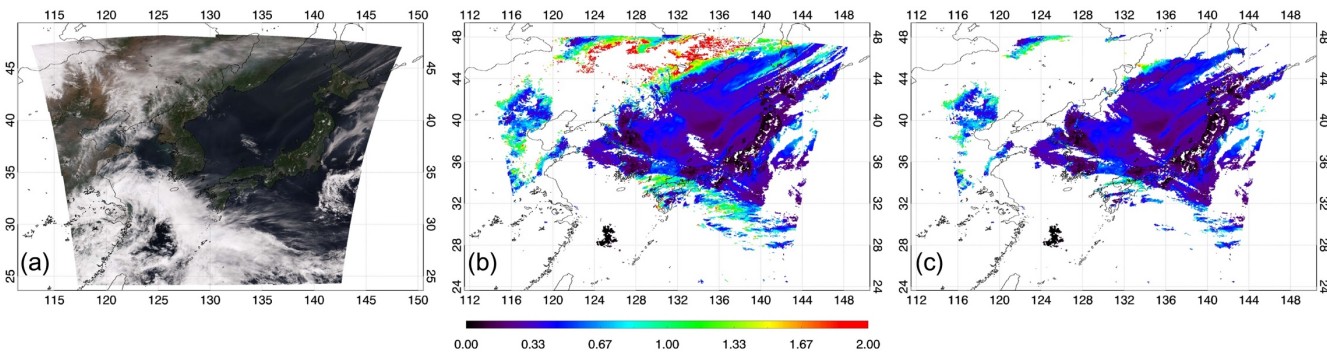

**Figure 1. Case of 22 May 2016, 13:30 LT (a) true color image, (b) GOCI YAER V2 AOD without additional cloud masking, and (c) GOCI YAER V2 AOD with additional cloud masking using AHI IR channels.**

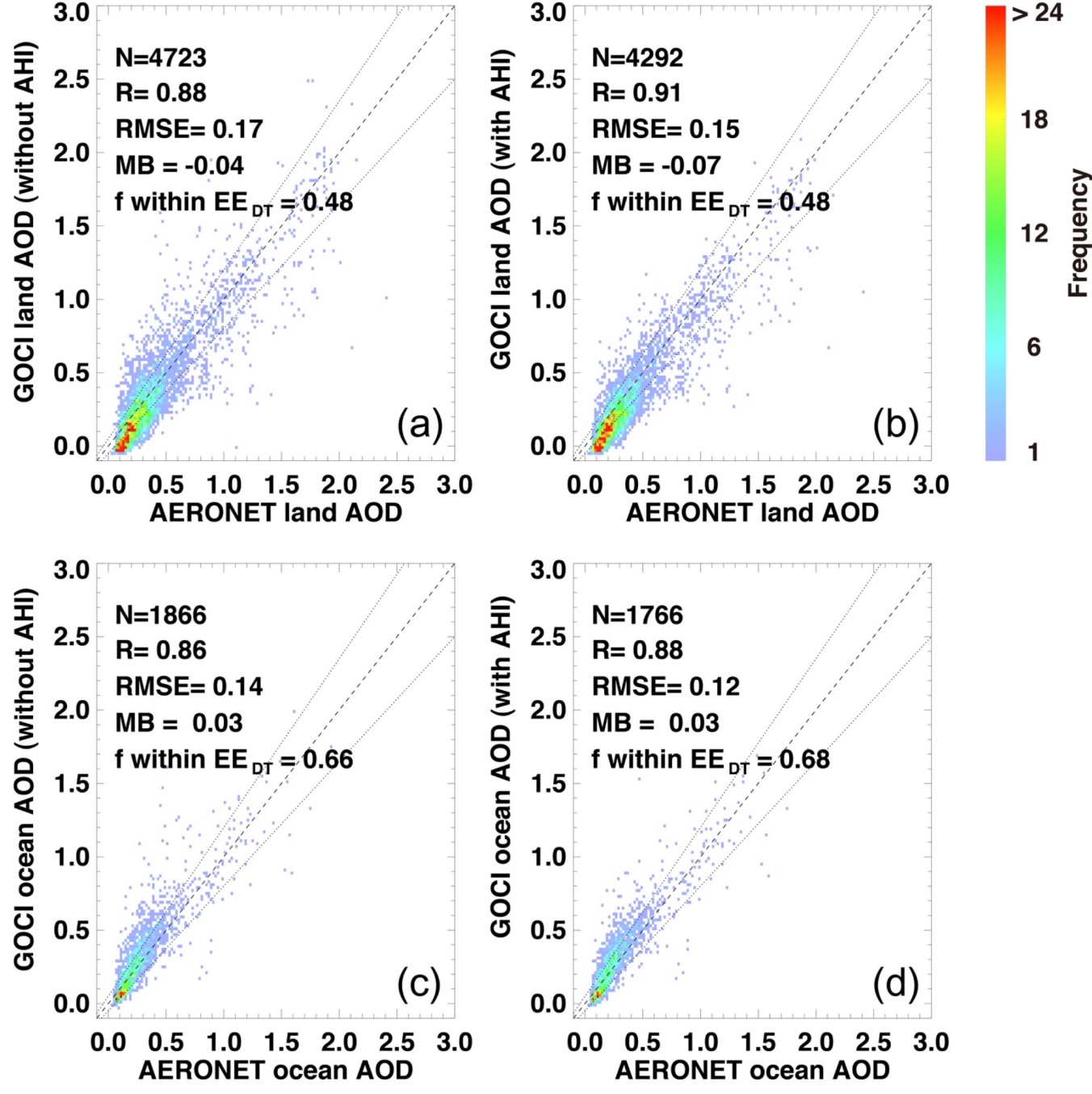

**Figure 2. Comparison of AERONET AOD and (a) GOCI land AOD without additional cloud masking, (b) GOCI land AOD with additional cloud masking, (a) GOCI ocean AOD without additional cloud masking, and (b) GOCI ocean AOD with additional cloud masking using AHI IR channels. Black lines indicate zero difference and the $EE_{DT}$ range $\pm(0.05 + 0.15 \times AOD_{AERONET})$.**

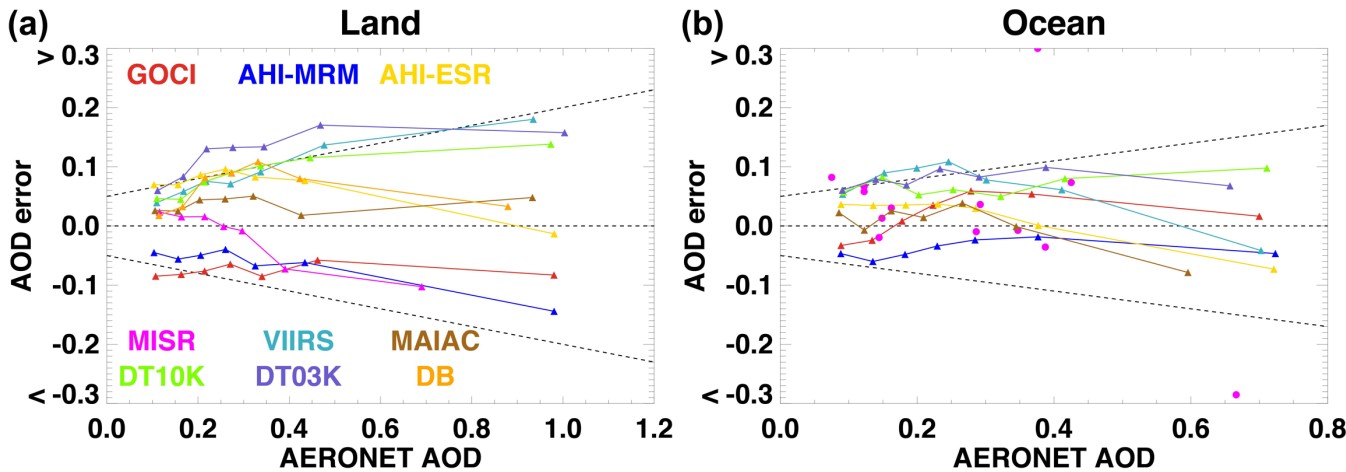

**Figure 3. Comparison of observed (a) land and (b) ocean AOD errors for AERONET AOD. For each product, the total collocated data are grouped into 7 bins according to AERONET AOD, except for MISR ocean AOD errors because of low collocation numbers. Each symbol indicates the median error for each collocated point, respectively. Black lines indicate zero difference and the EE$_{DT}$ range ±(0.05 + 0.15 × AOD$_{AERONET}$).**

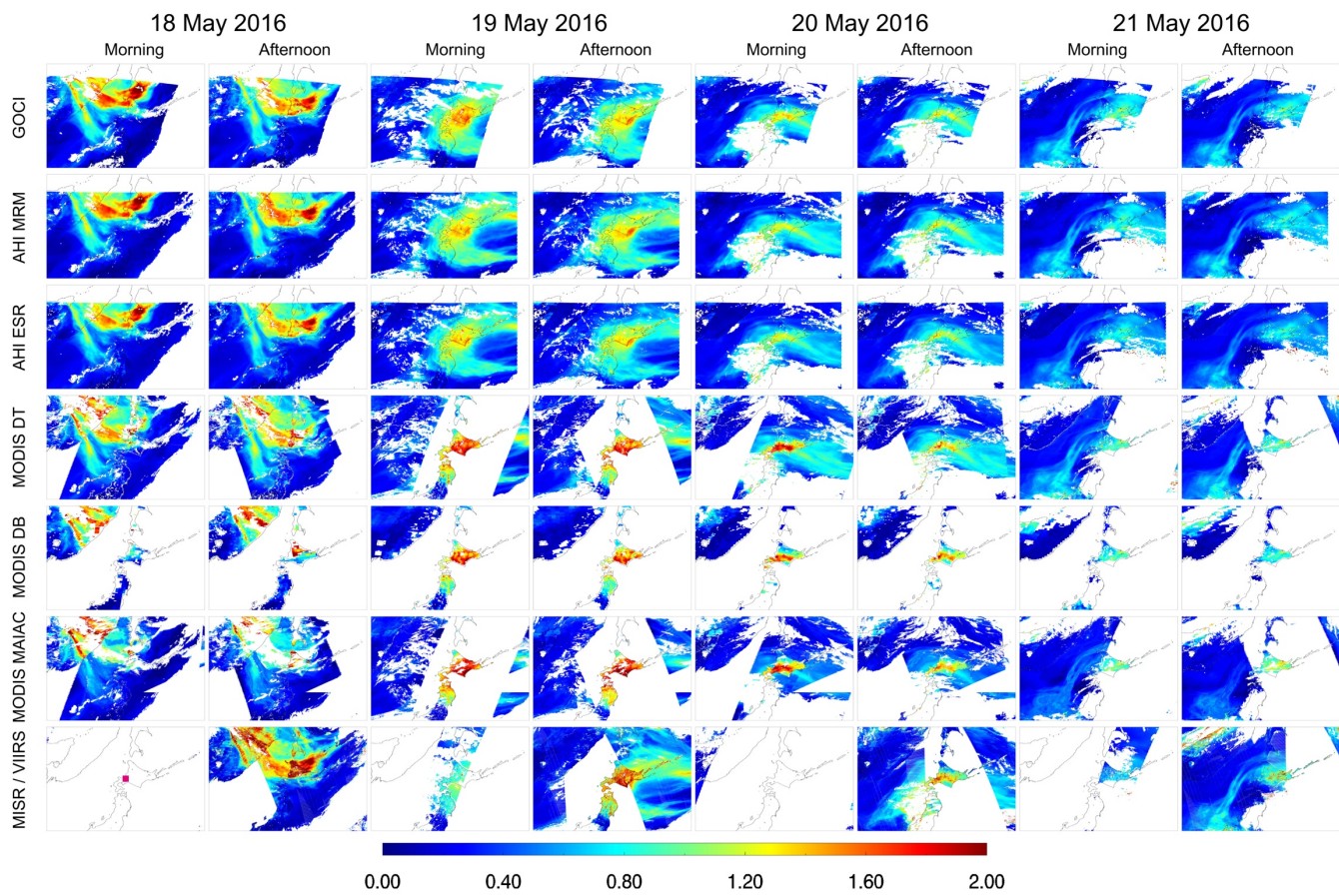

**Figure 4. AOD distributions from GOCI, AHI MRM, AHI ESR, MODIS DT03K, MODIS DB, MODIS MAIAC, MISR, and VIIRS over the Hokkaido region during 18–21 May 2016. Note that morning and afternoon AODs for GOCI and AHI refer to 10:30 and 13:30 LT, respectively, and for MODIS these refer to the Terra and Aqua measurements, respectively. MISR has only a morning measurement and VIIRS has only an afternoon measurement. A pink-color symbol in a left bottom panel presents a location of Hokkaido University AERONET site.**

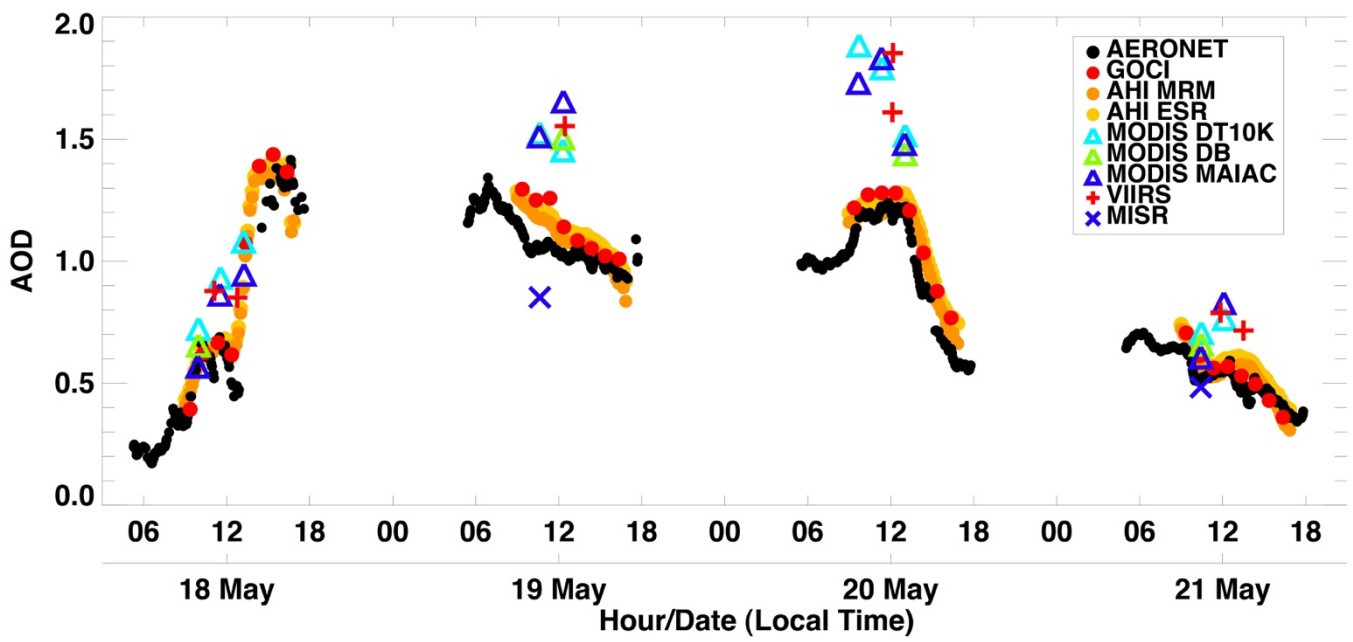

**Figure 5. Time series of multiple satellite AODs and AERONET AOD at the Hokkaido University site during 18–21 May 2016.**

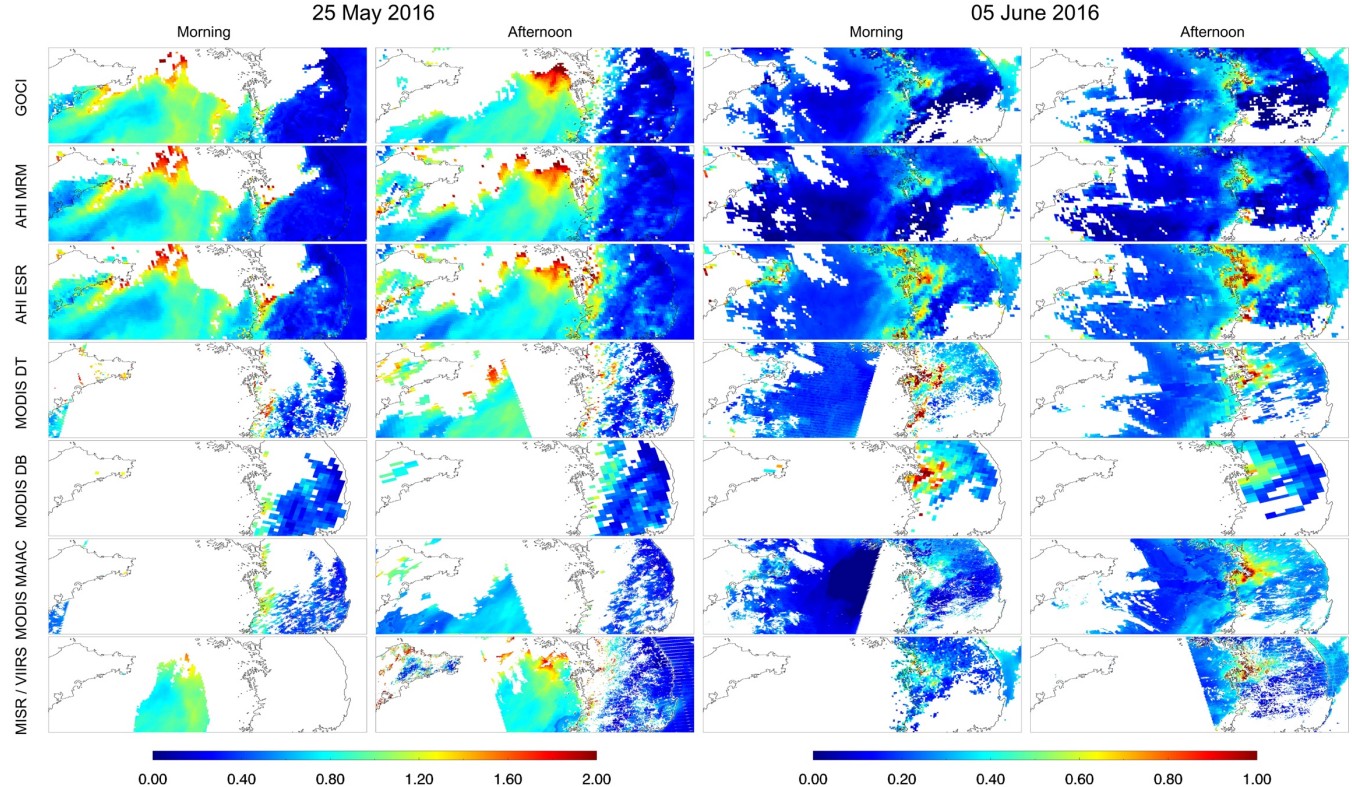

**Figure 6. AOD distributions from GOCI, AHI MRM, AHI ESR, MODIS DT03K, MODIS DB, MODIS MAIAC, MISR, and VIIRS over the Yellow Sea and Korean Peninsula (120°E–130°E and 35°N–38°N) at 25 May and 5 June 2016. Local time of morning and afternoon measurements is identical with Fig. 4.**

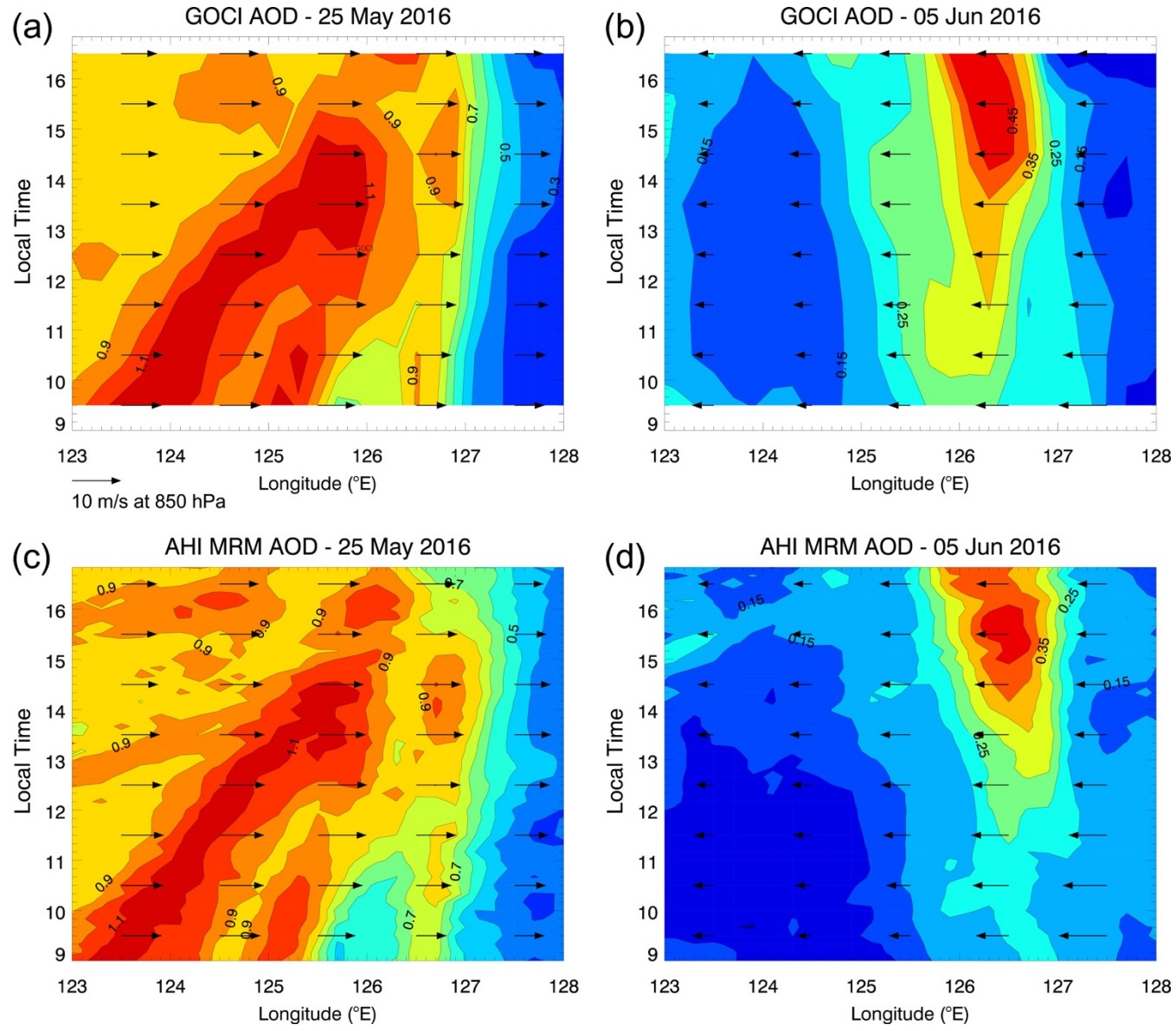

**Figure 7. Meridional mean GOCI AOD over the Yellow Sea and the Korean Peninsula (123°E–128°E and 35°N–38°N) at 0.2° longitude intervals on (a) 25 May and (b) 5 June 2016. Meridional mean AHI MRM AOD on (c) 25 May and (d) 5 June 2016. Overlapped arrows represent meridionally averaged zonal wind at 850 hPa.**

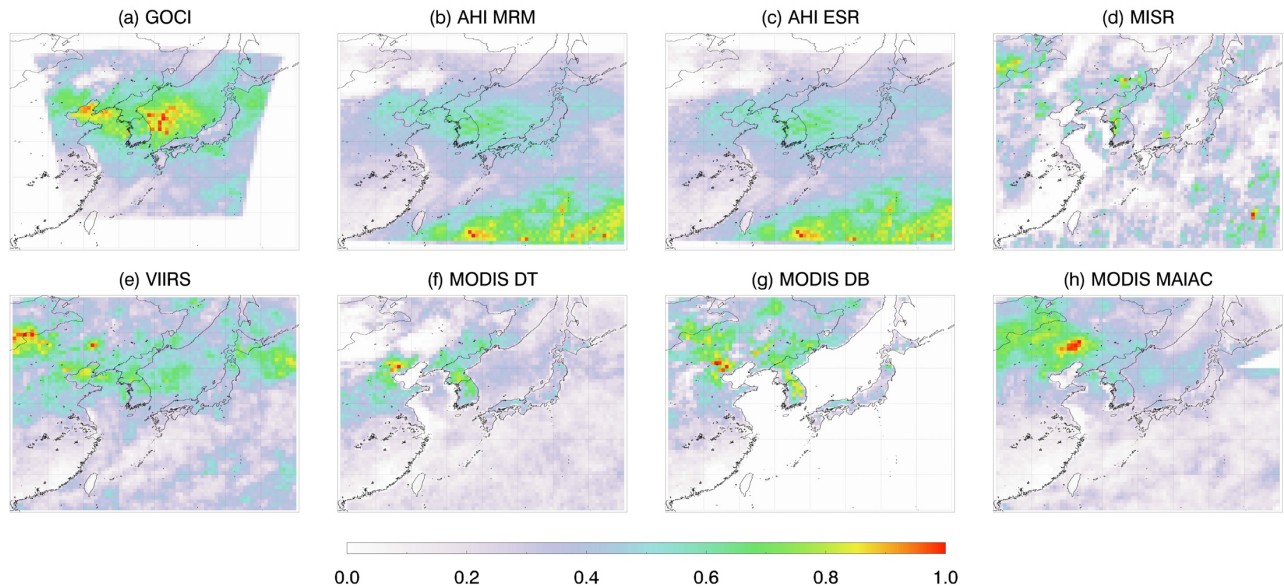

**Figure 8. Relative sampling frequency of L2 AOD pixels used to calculate mean AOD of (a) GOCI, (b) AHI MRM, (c) AHI ESR, (d) MISR, (e) VIIRS, (f) MODIS DT10K, (g) MODIS DB, and (h) MODIS MAIAC. The area of map is corresponding to 110–150°E and 20–50°N.**

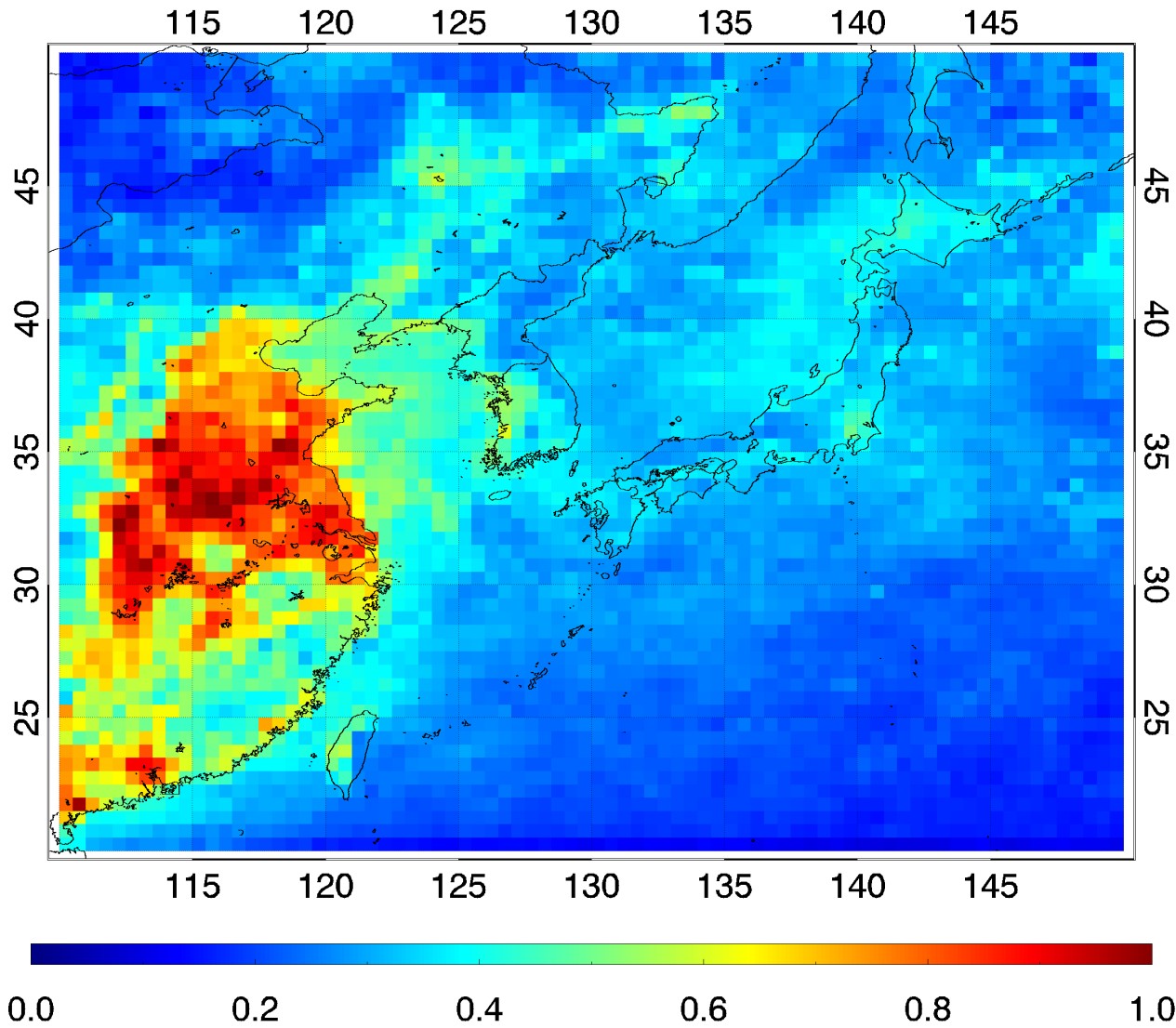

**Figure 9. Mean of daily fusion AOD (0.5° × 0.5° longitude–latitude) during the KORUS-AQ campaign period (1 May to 12 June 2016).**

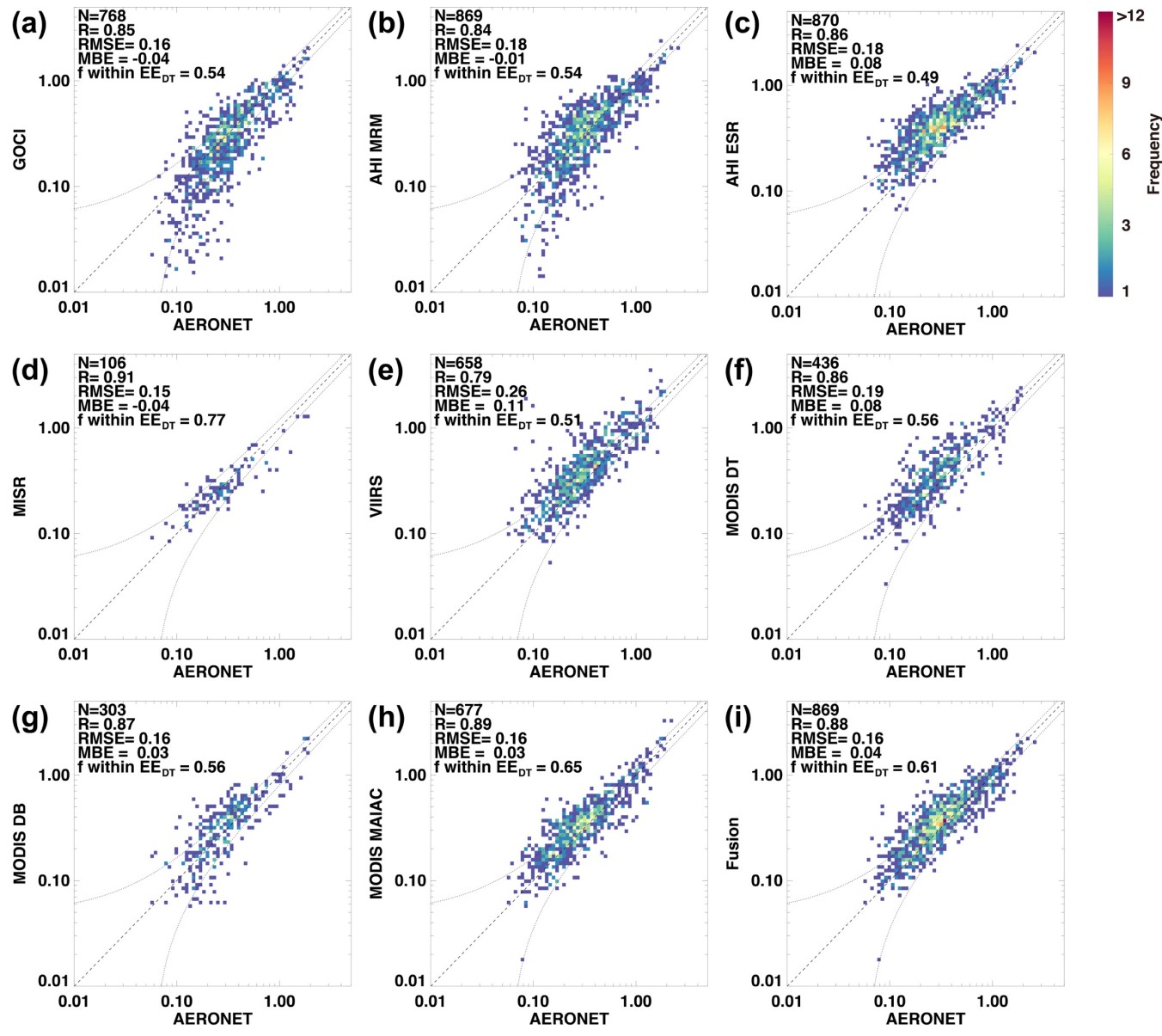

**Figure 10. Validation of daily-mean AOD using AERONET daily mean AOD during the KORUS-AQ campaign period (1 May to**
**12 June 2016) for the (a) GOCI, (b) AHI MRM, (c) AHI ESR, (d) MISR, (e) VIIRS, (f) MODIS DT10K, (g) MODIS DB, (h)**
**MODIS MAIAC, and (h) fusion products. Lines indicate the one-to-one line (dashed) and the range of EE$_{DT}$ (dotted).**

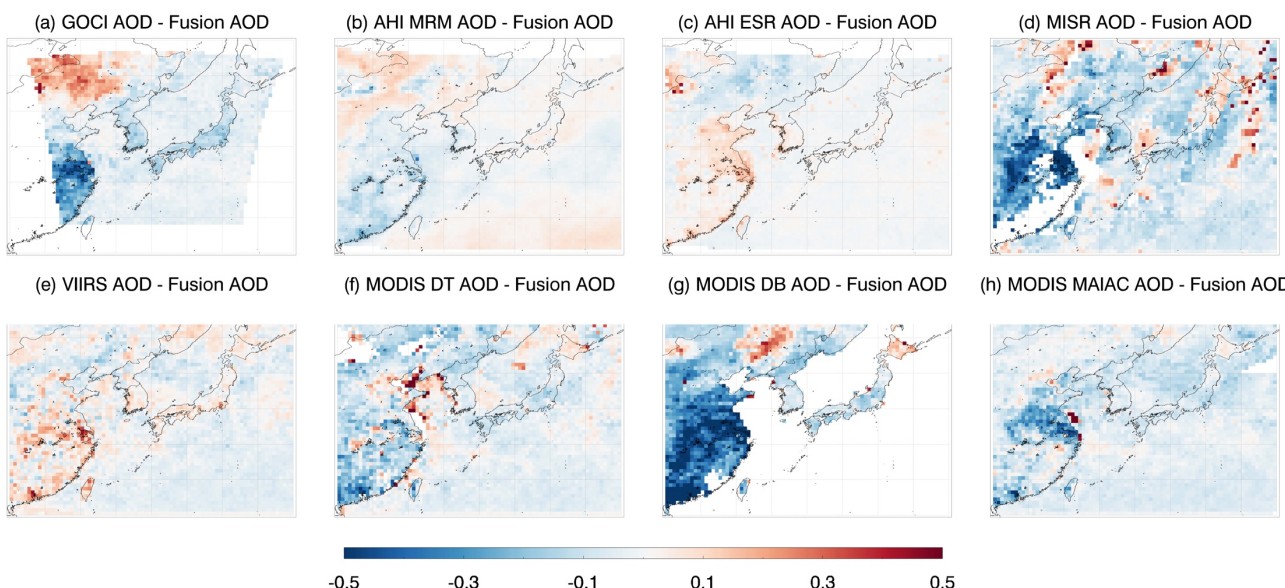

**Figure 11. Difference between campaign-period mean gridded AOD of (a) GOCI, (b) AHI MRM, (c) AHI ESR, (d) MISR, (e) VIIRS, (f) MODIS DT10K, (g) MODIS DB, and (h) MODIS MAIAC and mean of fusion AOD during the KORUS-AQ campaign period (1 May to 12 June 2016).**

