# Peer review of "Validation, comparison, and integration of GOCI, AHI, MODIS, MISR, and VIIRS aerosol optical depth over East Asia during the 2016 KORUS-AQ campaign"

_Atmospheric Measurement Techniques, 2019_

## Referee Comment (RC1) · Anonymous Referee #1 · 20 May 2019

This work describes validation results of aerosol retrievals from several polar-orbiting and geostationary sensors over the east Asia. The results indicate the accuracy of those aerosol products are quite good and similar. The content and topic of this manuscript is comprehensive, however, due to the wide coverage of different AOD retrievals from many satellites, the detail of the data selection and validation method is not fully described and some of the conclusions needs more discussion or proof to be reliable. Therefore, some major comments are suggested before it can be accepted.

1. Some revisions are necessary for the Introduction, the authors should add some de-

scription of the main concerns of this campaign, then link this goal with the experiments and analysis appeared in the following sections. 2. I was confused by the additional cloud masking used for GOCI Yonsei aerosol products. Since the goal of this study is to validate the aerosol properties from different satellites, the authors add additional cloud masking procedure to avoid the bias is not fair. Could you please discuss with this issue and give some description about how is the accuracy if no additional cloud masking is used. 3. It is surprising to find the number of AOD pixels (fig.7) for different sensors are of significant difference, the underlying reason is not fully discussed in Section 5.3. Besides, the impact of AOD sampling numbers on the analysis in fig.6 should be further discussed. 4. In Section 3.1, the authors discussed the indicators used to assess the performance of the AOD products, please add some necessary discussion on the criteria in using these indicators. 5. The references used for the description of the necessity of aerosol studies in the Introduction part is biased. 6. I suggest the author add some quantitative results of the validation in the Abstract. Please try to avoid sentences like "The AOD products analyzed here generally have high accuracy", which make no sense to the readers. 7. Why the authors only use the GOCI measurements in analyzing the case in Section 4.2? 8. Line5-10, the sentence "and high accuracy of other optical properties such as particle size or absorptivity beyond high accuracy of AOD to obtain more accurate ground level PM2.5 concentration and its species (Diner et al., 2018)" is difficult to understand. 9. The font of figure legends in fig.2 and 7 is too small, the quality of these figures should improve as well. 10. Please remove the lines between those points with no observations in figure 3

---

## Editor Comment (EC1) · Cheng Liu (Editor) · 21 May 2019

Review of the manuscript number AMT-2019-46 submitted to Atmospheric Measurement Techniques and entitled "Validation, comparison, and integration of GOCI, AHI, MODIS, MISR, and VIIRS aerosol optical depth over East Asia during the 2016 KORUS-AQ campaign" by Myungje Choi et al.

The paper describes the aerosol results and analysis from several different polar-orbiting and geostationary satellites. The author give a meaningful topic and we could

get a positive comparison response. It will be better that the authors could give more detailed outside validation using several ground-based instruments. Moreover, there should be a detailed description of data filter and validation method. I recommend publication after minor corrections.

Major comments: 1, The authors should have a detailed description for data filter of all the polar-orbiting and geostationary satellites. It will be meaningful if the authors could give the validation results of all the satellite analyzed in the manuscript calculated using a same algorithm. 2, In the manuscript, the authors also used AERONET data to validate the satellite results. It will be better that the authors could use other more ground-based instrument in different stations to validate and have a comparison with the satellites results. 3, From Figure2, we could find there are difference for the amount of data of different satellite. The authors should give detailed reasons (Due to errors or cloud?) 4, Why the authors only use the GOCI measurements in the case analysis (Section 4.2) Minor comments: Figure2 need to be improved. The font is too small that the readers can't see it clearly.
* * *

---

## Referee Comment (RC2) · Anonymous Referee #2 · 28 May 2019

The study validates and compares aerosol property, aerosol optical depth (AOD), of several satellites both GEO and LEO with AERONET or each satellite. The paper shows AOD accuracy of both GEO and LEO and indicates why the bias difference occurs. This study is useful to know what bias they have, to improve the retrieval algorithms and to select AOD data for air quality models. Below are my comments for the authors to consider before publishing the paper:

1. The paper mentions about using satellite combined AOD for air-quality model and mentions that an observation campaign in the paper also lead to improve air quality

model. Please added some sentences what problem of present models has (only for high time resolution?). Why combined AOD using GEO and LEO is useful (than that using only GEO)? If possible, please tell me the condition of data selection to make combine data because the authors validate and compared several GEO and LEO satellite data and also mention their bias. 2. In the abstract, the author has said that cloud screening is AOD difference between sensors. However, it is difficult to understand it from the paper. Please tell me why you did not meet the condition of each cloud screening when you compare AODs from several satellite. 3. Please add line between wavelength and the vertical line of MISR in Figure 1b (or remove lines between points). Please explain why MISR AOD accuracy is not good when MISR AOD is large over Land if you have some opinion. Please modify the figure because error bar is not clear. 4. Page 9, line17: "highly accurate", this expression is ambiguous. Please add what accuracy AODs are. In Page9: Does GOCI use Cox and Monk method over ocean? The AOD result of GOCI over ocean has positive bias, but in sentence the author have said "negative bias". Is it correct? 5. Figure 3. Please explain why MODIS DT and DB overestimate AOD. 6. Page10, line15∼: "high accuracy", this expression is ambiguous. Please explain why accurate it is, and why GEO results have continuous spatiotemporal distribution. 7. Figure 4, 5: I think that it is good to add some discussion including the wind speed. 8. Please correct to correct one. Page8 linr8-9, ". . . between 2011 and 2014 . . . negative bias 2015", page13 line29-30, ". . . period 2011-2015 . . . during the 2016"

---

## Author Comment (AC1) · 23 Jun 2019

We would like to thank the referee #1 for their constructive and useful comments. The supplement zip file contains the authors' responses to comments from the reviewer #1 and a revised manuscript.

Please also note the supplement to this comment:
https://www.atmos-meas-tech-discuss.net/amt-2019-46/amt-2019-46-AC1-supplement.zip

---

## Author Comment (AC2) · 23 Jun 2019

We would like to thank the referee #2 for their constructive and useful comments. The supplement zip file contains the authors' responses to comments from the reviewer #2and a revised manuscript.

Please also note the supplement to this comment:
https://www.atmos-meas-tech-discuss.net/amt-2019-46/amt-2019-46-AC2-supplement.zip

---

## Author Comment (AC3) · 23 Jun 2019

We would like to thank the editor for their constructive and useful comments. The supplement zip file contains the authors' responses to comments from the editor and a revised manuscript.

Please also note the supplement to this comment: https://www.atmos-meas-tech-discuss.net/amt-2019-46/amt-2019-46-AC3-supplement.zip